# Fitting large mixture models using stochastic component selection

## Abstract

Traditional methods for unsupervised learning of finite mixture models require to evaluate the likelihood of all components of the mixture. This becomes computationally prohibitive when the number of components is large, as it is, for example, in the sum-product (transform) networks. As a remedy, we propose an approach combining the expectation maximization and the Metropolis-Hastings algorithm to evaluate only a small number of, stochastically sampled, components, thus substantially reducing the computational cost. We put emphasis on generality of our method, equipping it with the ability to train both shallow and deep mixture models which involve complex, and possibly nonlinear, transformations. The performance of our method is illustrated in a variety of synthetic and real-data contexts, considering deep models, such as mixtures of normalizing flows and sum-product (transform) networks.

## 1  Introduction

Finite mixture models [40] constitute a fundamental class of density estimation models. They have been successfully applied in diverse fields, including bioinformatics [49], econometrics [10], engineering [33], etc. A mixture model relies on a weighted sum of probability distributions—here referred to as *components*—to cluster $N$ unlabelled datapoints into $K$ categories. The traditional maximum likelihood techniques train the model by optimizing either (i) the marginal likelihood via gradient-descent [50] or (ii) the evidence lower bound via variational methods [4], including the expectation-maximization (EM) [13]. The dependence structure among approximate, variational, distributions then ranges from the fully independent (mean-field) [25] to fully dependent [30]. The sampling-based techniques target the posterior distribution using sequential Monte Carlo [9] or Markov chain Monte Carlo [52], e.g. via the Gibbs [34] or Metropolis-Hastings sampling [38]. The computational cost of these methods typically scales with $\mathcal{O}(TKND)$ operations, where $N$ and $K$ are defined above, $T$ is the number of iterations and $D$ is the dimension of data.

Various methods to decrease the computational cost via any factor in $\mathcal{O}(TKND)$ have been proposed. $T$ can be lowered by proper initialization, e.g. the optimal seeding [5]; an efficient step-size schedule, e.g. the line-search [58]; or increased estimation precision, e.g. the variance reduction [8]. $N$ is often reduced using the coreset methods, which approximate the original dataset by a weighted dataset such that the exact and approximate marginal likelihoods are close. The weighted variants of the variational [17, 59, 6] and sampling-based [39] methods then process the coresets. Reducing $D$ relies on the compression of data into smaller representations via random projections [53, 2], which is achieved in two ways: (i) each data item is projected into an individual representation [11]; (ii) all data items are projected into an overall representation, commonly referred to as sketch [28, 22].

Nevertheless, all the aforementioned techniques—including those with reduced computational cost— evaluate all $K$ components. This is very demanding for large models, and the problem is even more

Table 1: The computational features of various EM algorithms. We compare whether the methods (i) perform the computations with a reduced number of data (minibatching), (ii) update a lower number of statistics, (iii) make less evaluations of the conditional likelihood, and (iv) are suitable for training of deep models. Here, EM, SA, S, T, MC and MH stand for expectation-maximization, stochastic approximation, sparse, truncated, Monte Carlo and Metropolis-Hastings, respectively.

| Feature / Algorithm | EM [13] | SAEM [44] | SSAEM [24] | TSAEM [18] | MCSAEM [1] | MHSAEM (ours) |
|---|---|---|---|---|---|---|
| $B < N$ datapoints | ✗ | ✓ | ✓ | ✓ | ✓ | ✓ |
| $M < K$ statistics | ✗ | ✗ | ✓ | ✓ | ✓ | ✓ |
| $M < K$ likelihoods | ✗ | ✗ | ✗ | ✓ | ✗ | ✓ |
| deep models | ✗ | ✗ | ✗ | ✗ | ✗ | ✓ |

severe for mixtures involving intricate models, such as neural networks [21, 42], Gaussian processes [57], normalizing flows [48]; or deep mixtures, including sum-product (transform) networks [45, 47], deep Gaussian mixture models [55], etc. In spite of this, a little attention has been paid to the design of algorithms which does not evaluate all $K$ components. The notable exceptions are the sparse EM algorithm [24] and the truncated variational EM algorithm [18], see Table 1 and Section 5 for details. Moreover, the methods are mostly tailored for a specific class of mixture models, e.g. the Gaussian mixture models.

In this paper, we make the following contributions:

- We propose an EM-based algorithm which relies on the MH sampler to stochastically evaluate less components in mixture models, substantially reducing the computational cost.

- We design our method to enable optimization of fairly generic EM objective functions, making it suitable for training of both shallow and deep mixture models.

- We apply our approach to Gaussian mixture mdoels (GMMs) and their generalizations: sum-product-transform networks (SPTNs) and mixtures of real-valued non-volume preserving (real NVP) flows [15], reaching approximately $100\times$ speed-up compared to state-of-the-art methods.

## 2  Problem formulation

A finite mixture model characterizes the relation between an observed (known) variable, $x \in \mathsf{X} \subseteq \mathbb{R}^D$, and a latent (unknown) variable, $z \in \mathsf{Z} := \{1, \ldots, K\}$, via the marginal (incomplete-data) likelihood in the following form:

$$p_\theta(x) = \sum_{k=1}^{K} p_{\eta_k}(x|z = k)p_{\pi_k}(z = k), \tag{1}$$

where $\theta := (\pi_1, \eta_1, \ldots, \pi_K, \eta_K) \in \Theta$ are unknown parameters. Here, $\eta_z$ are the parameters of the conditional likelihood, $p_{\eta_z}(x|z)$, and $\pi_z$ is the weight which parameterizes the prior, $p_{\pi_z}(z) = \pi_z$, and satisfies $0 \leq \pi_k \leq 1$ for each $k \in \mathsf{Z}$ and $\sum_{k=1}^{K} \pi_k = 1$.

Given a set of independent and identically distributed data, $\mathbf{x} := (x_i)_{i=1}^{N}$, our goal is to learn the unknown parameters of the marginal log-likelihood,

$$\mathcal{L}(\theta) := \log p_\theta(\mathbf{x}) = \sum_{i=1}^{N} \log \sum_{k=1}^{K} p_{\eta_k}(x_i|z_i = k)p_{\pi_k}(z_i = k). \tag{2}$$

The marginalization in (2) is tractable for almost all forms of $p_{\eta_z}(x|z)$. Indeed, we consider $p_{\eta_z}(x|z)$ to belong to an arbitrary family of $\eta_z$-differentiable probability distributions. However, we assume that $K$ is high, making the marginalization in (2) computationally costly, thus rendering the optimization objective presumably intractable. Therefore, we want to design a computationally efficient algorithm, requiring only $M < K$ evaluations of $p_{\eta_z}(x|z)$ at each iteration.

## 3 The EM algorithm

The maximum likelihood estimation seeks the parameters maximizing the marginal log-likelihood, $\theta^{ML} := \arg\max_{\theta \in \Theta} \mathcal{L}(\theta)$. The traditional EM algorithm [13] addresses this task indirectly, i.e. by optimizing the evidence lower bound (ELBO),

$$\mathcal{L}(\theta) \geq \mathcal{Q}(\theta) + \mathcal{H}(\hat{\theta}) := \text{ELBO}(\hat{\theta}), \tag{3}$$

where $\mathcal{H}(\hat{\theta}) := -\mathsf{E}_{p_{\hat{\theta}}(\mathbf{z}|\mathbf{x})}[\log p_{\hat{\theta}}(\mathbf{z}|\mathbf{x})]$ is the differential entropy at an estimate, $\hat{\theta} \in \Theta$, and

$$\mathcal{Q}(\theta) := \mathsf{E}_{p_{\hat{\theta}}(\mathbf{z}|\mathbf{x})}[\log p_\theta(\mathbf{z}, \mathbf{x})] = \sum_{i=1}^{N} \sum_{k=1}^{K} p_\theta(z_i = k | x_i) \log p_\theta(z_i = k, x_i) \tag{4}$$

is the EM objective function. Here, $p_\theta(\mathbf{z}, \mathbf{x})$ is the joint (complete-data) likelihood, and $p_\theta(\mathbf{z}|\mathbf{x})$ is the posterior distribution over the latent variables $\mathbf{z} := (z_i)_{i=1}^{N}$. Given an initial value, $\theta_0$, the EM algorithm produces a sequence of estimates, $(\theta_t)_{t=1}^{T}$, by alternating between the expectation (E) and maximization (M) steps,

$$\text{E-step:} \quad \mathcal{Q}_{t-1}(\theta), \tag{5}$$

$$\text{M-step:} \quad \theta_t := \arg\max_{\theta \in \Theta} \mathcal{Q}_{t-1}(\theta). \tag{6}$$

This sequence is guaranteed to monotonically tighten the ELBO, arriving at a local optimum of (2) under mild regularity assumptions [56].

The EM algorithm is computationally expensive, since (4) evaluates $p_\theta(z_i, x_i)$ for each $z_i \in \mathsf{Z}$ and $i \in (1, \ldots, N)$. This has to be performed for all $t \in (1, \ldots, T)$ in (5). Albeit the marginal factor, $p_{\pi_z}(z)$, is just the cheap categorical distribution, the conditional factor, $p_{\eta_z}(x|z)$, typically involves high-dimensional operations (e.g., the inversion of the full $D \times D$-dimensional covariance matrices in the GMMs). Moreover, the M-step (6) is also expensive for large $K$. This holds despite that (6) can be reduced to closed-form updates of expected sufficient statistics for $p_{\eta_z}(x|z)$ belonging to the exponential family [44] (again, due to high $D$). All in all, the computational complexity of the EM algorithm scales with $\mathcal{O}(TDNK)$.

If (6) cannot be computed under a closed-form solution, one can resort to direct gradient-descent optimization of $\mathcal{Q}(\theta)$, where $\arg\max$ is replaced by one (or more) step(s) of a gradient descent technique. The EM algorithm is then referred to as the generalized EM algorithm [56].

## 4 The generalized MHSAEM algorithm

We design a version of the generalized EM algorithm suitable for scenarios where (4) can represent deep, discrete, latent variable models, thus being parameterized by possibly complex nonlinear transformations. We particularly focus on decreasing the the number of operations in the generalized EM algorithm from $\mathcal{O}(TDNK)$ to $\mathcal{O}(TDBM)$, where $B \ll N$ and $M \ll K$.

### 4.1 E-step

We reduce the cost of evaluating the EM objective function (4) by combining the minibatching (as used many times before) and the Monte Carlo sampling. Namely, the specific application of the latter to generic mixture models is the key contribution of this paper.

*Minibatching.* At each iteration, $t$, we compute the conditional expectation in (4) only for a subset— here referred to as a *minibatch*—of the original full dataset, i.e. $(x_i)_{i \in I}$. Here, $I$ is a set of $B \ll N$ indices, $i$, sampled uniformly without replacement from $(1, \ldots, N)$. This substantially decreases the necessary computations compared to the full sweep over all $N$ datapoints [23].

*Monte Carlo sampling.* For each $i \in I$, we want to draw $M \ll K$ random samples from $p_\theta(z_i|x_i)$ in order to obtain a Monte Carlo estimate of (4). The straightforward way to do this would be to draw the samples directly from $p_\theta(z_i|x_i)$. However, direct sampling from $p_\theta(z_i|x_i)$ does not lead to any substantial decrease in the number of operations. This is caused by the fact that even for a *single* sample of $z_i$, we have to first compute the normalizing factor, $p_\theta(x_i)$, to obtaining the posterior,

$p_\theta(z_i|x_i)$. This requires $K$ expensive evaluations of $p_\theta(z_i, x_i)$, which is precisely what we want to avoid. Our approach is to resort to the Markov chain Monte Carlo (MCMC), which allows us to sample from $p_\theta(z_i|x_i)$, with the computational complexity decreasing to only a *single* evaluation of $p_\theta(z_i, x_i)$ per a *single* sample of $z_i$.

MCMC methods obviate the computation of the normalizing factor in $p_\theta(z_i|x_i)$ by simulating a Markov chain, $(z_{i,t})_{t=1}^T$, from a transition kernel, $z_{i,t} \sim P(z_{i,t-1}, \cdot)$, which leaves $p_\theta(z_i|x_i)$ as its unique stationary (invariant) distribution, starting from an initial value $z_{i,0}$. The specific form of $P$ determines the structure of an MCMC method. We chose the Metropolis-Hastings (MH) sampler, which represents $P(z_{i,t-1}, z_{i,t})$ as follows: given $\bar{z}_i := z_{i,t-1}$, draw a sample from the *proposal* distribution $z_i \sim q(\cdot|\bar{z}_i)$, compute the acceptance ratio,

$$\alpha(\bar{z}_i, z_i) := \min\left\{1, \frac{p_{\eta_{z_i,t-1}}(x_i|z_i)\pi_{z_i,t-1}q(\bar{z}_i|z_i)}{p_{\eta_{\bar{z}_i,t-1}}(x_i|\bar{z}_i)\pi_{\bar{z}_i,t-1}q(z_i|\bar{z}_i)}\right\}, \tag{7}$$

and, if $u < \alpha(\bar{z}_i, z_i)$—where $u$ is drawn from a uniform distribution, Uniform$(0, 1)$—accept the sample and set $z_{i,t} = z_i$; otherwise, set $z_{i,t} = \bar{z}_i$. For each $i \in I$ and $t \in (1, \ldots, T)$, we repeat this process $M$ times, construing a set $\mathbf{z}_{i,t} = (z_{i,t}^1, \ldots, z_{i,t}^M)$. Therefore, at every current iteration, $t$, we continue to extend the chain from the point where we left at the previous iteration, $t-1$, by taking $\bar{z}_i = z_{i,t-1}^M$. Under mild regularity assumptions [52], the chain passes the transition period (the burn-in phase), and the samples can then be used to approximate the conditional expectation in (4) as follows:

$$\widehat{\mathcal{Q}}_{t-1}(\theta) = \frac{1}{M} \sum_{i \in I} \sum_{z \in \mathbf{z}_{i,t}} \log p_{\eta_z}(x_i|z)\pi_z. \tag{8}$$

Note that, to ensure this approach is truly efficient, we have to draw only $M \ll K$ samples at each iteration, $t$; otherwise, for $M \approx K$, we may rather compute the exact marginalization in (4), since it is tractable (but computationally costly).

## 4.2 M-step

Assume for a moment that (6) with $\mathcal{Q}_{t-1}(\theta)$ given by (8) has a closed-form solution, yielding an estimate of $\theta$. Such an estimate would have a high variance, converging only for $M \to \infty$ and $T \to \infty$ [19]. The main reason is that the samples would not be reused over the iterations, $t$, thus wasting computational resources. We consider that there is no closed-form solution of (6), and—to ensure that the samples (and thus computations) are recycled over the iterations—we use the stochastic approximation (SA) [51] to optimize (8). This is analogous to applying a stochastic gradient-descent method, $\theta_t = \theta_{t-1} + \gamma_t \nabla_\theta \widehat{\mathcal{Q}}_{t-1}(\theta)$, where $\gamma_t$ is the step-size, satisfying the Robbins-Monro constraints, $\gamma_t \in [0, 1]$, $\sum_{t \geq 1} \gamma_t = \infty$, $\sum_{t \geq 1} \gamma_t^2 < \infty$, and $\nabla_\theta$ is the gradient w.r.t. $\theta$. In this way, the computations made in $\nabla_\theta \widehat{\mathcal{Q}}$ are accumulated via $\theta_t$ and reused over the iterations.

The parameters $\eta_z$ have a different form based on a specific case of $p_{\eta_z}(x|z)$, whereas $\pi_z$ is a permanent structure in (1). Therefore, without loss of generality, we split (6) into a generic part and a fixed part as follows:

$$\eta_{k,t} = \eta_{k,t-1} + \gamma_t \nabla_{\eta_k} \widehat{\mathcal{Q}}_{t-1}(\theta), \tag{9a}$$

$$\nu_{k,t} = \nu_{k,t-1} + \gamma_t \nabla_{\nu_k} \widehat{\mathcal{Q}}_{t-1}(\theta), \tag{9b}$$

where—to ensure that the probabilities, $(\pi_{k,t})_{k=1}^K$, satisfy the constraints (Section 2)—we transform $\nabla_{\pi_k} \widehat{\mathcal{Q}}$ via $\nu_k = \log \pi_k$ and optimize w.r.t. $\nu_k$. Then, to obtain $(\pi_{k,t})_{k=1}^K$ from $\boldsymbol{\nu}_t := (\nu_{k,t})_{k=1}^K$, we use the softmax function, i.e. $\pi_{k,t} := \text{softmax}(\boldsymbol{\nu}_t)_k := \exp(\nu_{k,t})/\sum_{l=1}^K \exp(\nu_{l,t})$.

Computing the gradients for all pairs of $(\nu_k, \eta_k)_{k=1}^K$ would be inefficient, especially since $\mathbf{z}_{i,t}$ contains only a small number of unique values of $Z$ for $M \ll K$. Consequently, we compute $\nabla_{\eta_k} \widehat{\mathcal{Q}}$ and $\nabla_{\nu_k} \widehat{\mathcal{Q}}$ only for $k \in \text{unique}(\mathbf{z}_{i,t})$. We summarize the proposed approach in Algorithm 1.

## 4.3 Proposal distribution

The choice of the proposal distribution has a significant impact on the speed of convergence and the computational cost of the proposed algorithm. Here, we discuss various possible choices of $q(z_i|\bar{z}_i)$.

---

**Algorithm 1** The generalized MHSAEM algorithm

---
**Input:** $\theta_0, (\mathbf{z}_{i,0})_{i=1}^N, (\mathbf{x}_i)_{i=1}^N$
**Output:** $(\theta_t)_{t=1}^T$
  **for** $t \in (1, \ldots, T)$ or until convergence **do**
     form the set $I = (i_j)_{j=1}^B$ by sampling (without replacement) $B$ indices $i \sim (1, \ldots, N)$
    **for** $i \in I$ **do**
       set $\bar{z}_i$ as the last element of $\mathbf{z}_{i,t-1}$
       **for** $j \in (1, \ldots, M)$ **do**
          sample $z_i \sim q(z_i | \bar{z}_i)$
          sample $u \sim \text{Uniform}(0, 1)$
          compute $\alpha(\bar{z}_i, z_i)$ in (7)
          **if** $u < \alpha(\bar{z}_i, z_i)$ **then**
             set $z_{i,t}^j = z_i$ and $\bar{z}_i = z_i$
          **else**
             set $z_{i,t}^j = \bar{z}_i$
          **end if**
       **end for**
       set $\mathbf{z}_{i,t} = (z_{i,t}^1, \ldots, z_{i,t}^M)$
    **end for**
    compute (8)
    compute (9) for $k \in \text{unique}(\mathbf{z}_{i,t})$
    compute $\pi_{k,t} := \text{softmax}(\boldsymbol{\nu}_t)_k$ for $k \in \mathsf{Z}$
  **end for**

---

*Optimal proposal (O).* The optimal proposal distribution is $q(z_i | \bar{z}_i) := q(z_i) := p_\theta(z_i | x_i)$. This ensures that the acceptance rate (7) is always $\alpha(\bar{z}_i, z_i) = 1$. However, the need to perform $K$ expensive evaluations of $p_\theta(z_i, x_i)$ before sampling from $p_\theta(z_i | x_i)$ is the reason we resorted to the MH sampler in the first place. We consider this case only to set the upper limit on admissible computational cost and to study the impact of sub-optimal proposal distribtions.

*Uniform proposal (U).* The uniform distribution on the discrete interval from 1 to $K$, i.e. $q(z_i | \bar{z}_i) := q(z_i) := \text{Uniform}(1, K)$, is the simplest and computationally cheapest variant of the proposal distribution. However, due to poor mixing properties, the algorithm may converge slowly for high $K$.

*Tabular proposal with forgetting (TF).* The key requirement to design a proposal distribution is to restrict its computational complexity somewhere between that of the U and O proposals. One way to satisfy this constraint is to use the Markov chain, $(z_{i,t})_{t=1}^T$, to learn a transition kernel, $p(z_i | \bar{z}_i)$, see, e.g. [3]. Unfortunately, this would require us to store a table with $K^2$ entries for each $i \in (1, \ldots, N)$, which is very demanding even for moderate $K$ and $N$. Therefore, we break the dependence in the Markov chain and define: $q(z_i | \bar{z}_i) := q_{\boldsymbol{\alpha}_i}(z_i) := \mathcal{C}(\boldsymbol{\alpha}_i)$, where $\mathcal{C}(\boldsymbol{\alpha}_i) \propto \Pi_{k=1}^K \alpha_{k,i}^{\mathbb{1}(z_i = k)}$ is the categorical distribution with the weights $\boldsymbol{\alpha}_i := (\alpha_{1,i}, \ldots, \alpha_{K,i})$. For $\mathcal{L}(\boldsymbol{\alpha}_i) := \Sigma_{\tau=1}^t \log q_{\boldsymbol{\alpha}_i}(z_{i,\tau})$, we obtain an estimate of $\boldsymbol{\alpha}_i$ at iteration $t$ as follows: $\boldsymbol{\alpha}_{i,t} := \arg\max_{\boldsymbol{\alpha}_i} \mathcal{L}(\boldsymbol{\alpha}_i) = \frac{n_{i,t}}{t}$, with $n_{i,t} = \Sigma_{\tau=1}^t \mathbf{e}_{z_{i,t}}$, where $\mathbf{e}_k$ is the standard basis vector (a one-hot vector) with one at $k$th position and zeros otherwise. This can be further rewritten into a recursive form: $n_{i,t} = n_{i,t-1} + \mathbf{e}_{z_{i,t}}$ or, using the Robbins-Monro step-size, $n_{i,t} = (\mathbf{1} - \mathbf{e}_{z_{i,t}} \gamma_t) \odot n_{i,t-1} + \gamma_t \mathbf{e}_{z_{i,t}}$, where $\mathbf{1}$ is the vector of ones, and $\odot$ is the Hadamard product. We refer to this case simply as "table with forgetting" (TF) due to that it represents $N \times K$ table in the memory and $\gamma_t$ is a forgetting factor.

## 5   Related work

*Stochastic approximation expectation-maximization.* The application of SA to prevent the evaluation of all $K$ components in mixture models has been overlooked for a long time. The reason is that the original motivation to combine the EM algorithm with SA is to address the analytical intractability of the expected value under $p_\theta(z | x)$ in (4), which is, however, almost always tractable for mixture models. The intractability issue is addressed by either the Monte Carlo SAEM (MCSAEM) [12] or the Markov chain Monte Carlo SAEM (MCMCSAEM) [31]. Applying the former approach to mixture models would be inefficient, since it evaluates $K$ joint distributions, $p_\theta(z, x)$, before drawing

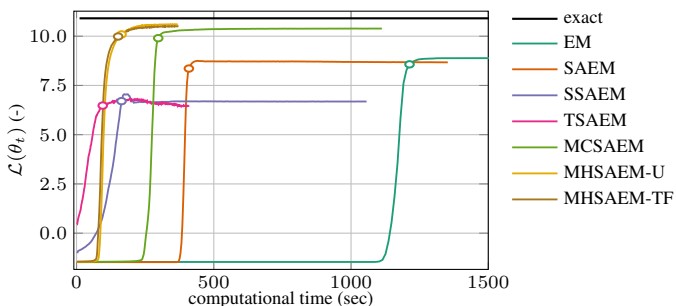

Figure 1: The training log-likelihood, $\mathcal{L}(\theta_t)$, versus the computational time (in seconds). Here, on the x-axis, the computational time at a current iteration, $t$, is obtained by accumulating the time from the previous iterations. $\circ$ corresponds to $\mathcal{L}(\theta_{t_{95}})$, where $t_{95}$ is the iteration of reaching 95% of $\max \mathcal{L}(\theta_t)$. The projection of $\circ$ on the x-axis gives the time to reach $\mathcal{L}(\theta_{t_{95}})$. This experiment was performed with the following settings: $(D, K, N, \omega, B, M, T) = (10, 100, 10k, 0.1, 200, 2, 20k)$, see Section 6.1 for details. The results are averaged over five repetitions.

M samples from $p_\theta(z|x)$. Therefore, this method reduces only the computational cost of updating the *sufficient statistics*. This is addressed by the latter approach, where $M < K$ samples from a proposal distribution, $q(z|x)$, is used to calculate $p_\theta(z,x)$ and also the sufficient statistics. However, all these methods process all data at every iteration, providing only a limited advantage over the conventional EM algorithm. Minibatch versions of these techniques have recently been proposed [27, 32, 1].

All the above methods commonly assume $p_\theta(z,x)$ belonging to the exponential family. This provides a convenient, but limiting, property which allows (6) to be computed under a closed-form solution. The main contribution of our work is to release this restrictive assumption by admitting that $p_\theta(z,x)$ (and thus $\mathcal{Q}$) is given by possibly complex and intractable transformations.

*Sparse and truncated variational techniques.* There is only a small body of methods explicitly reducing the number of evaluated components. Their common aspect is that they follow from the variational framework, where the exact posterior, $p_\theta(z|x)$, is approximated by a variational posterior, $q(z|x)$. This sparse, approximate, posterior is defined over a lower number of components, $M \ll K$, such that only the important components are selected, relying on relaxation of the hard EM algorithm from taking a single $M = 1$ assignment [26] to taking multiple $M \ll K$ assignments. The sparse SAEM (SSAEM) algorithm [24] selects the components by a quick partial sorting of the posterior probabilities, $p_\theta(z|x)$. Again, this requires $K$ evaluations of $p_\theta(z,x)$ before the sorting, thus only reducing the amount of updated statistics. Similarly, the truncated SAEM (TSAEM) algorithm [18] selects $M < K$ cluster-to-cluster and $\bar{M} < K$ cluster-to-datapoint minimal Euclidean distances, preventing the problem in the SSAEM algorithm. However, all these distances are evaluated for all components in a pairwise manner, leading to $K^2$-computational complexity, which makes the saving dubious. Similarly as before, these methods assume $p_\theta(z,x)$ to belong to the exponential family.

We summarize the distinguishing features of the above discussed methods in Table 1.

# 6 Experiments

To demonstrate the key features of our algorithm—its low computational complexity, competitive learning performance, and generality—we use it below to train: (i) GMMs on synthetic datasets, and (ii) SPTNs [47] and (iii) mixtures of real NVP flows [48] on real datasets. All experiments have been performed on a Slurm cluster equipped with Intel Xeon Scalable Gold 6146 with 384GB of RAM.

## 6.1 Gaussian mixture models

Consider the special case of a data-generating distribution given by (1), with the components taking the form of the multivariate Gaussian distribution, $p_{\eta_z}(x|z) = \mathcal{N}(x; \mu_z, \Sigma_z)$, where $\mu_z$ is the mean value and $\Sigma_z$ is the covariance matrix. The difficulty of learning GMMs heavily depends on the degree of interaction among all mixture components, hence having the ability to generate synthetic datasets with arbitrary overlap characteristics between all pairs of components is crucial for systematic

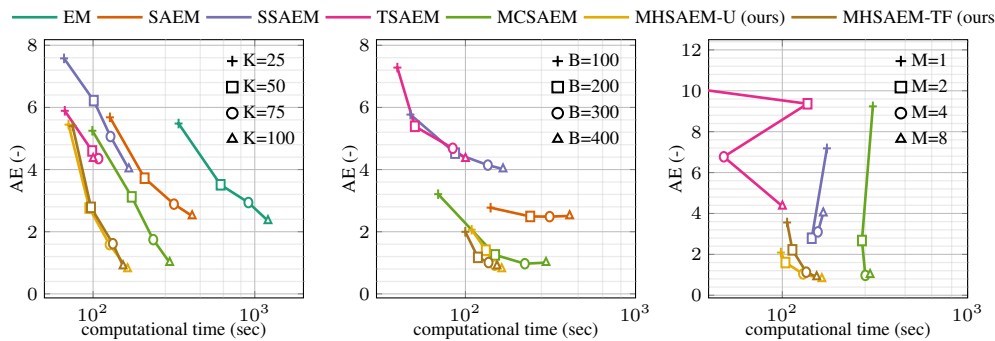

Figure 2: The absolute error, $AE = |\mathcal{L}(\theta_{t_{95}}) - \mathcal{L}(\theta)|$, versus the computational time (in seconds). All experiments use the following settings: $(D, K, N, \omega, B, M, T) = (10, 100, 10k, 0.1, 200, 2, 20k)$, where the number of components, $K$, (left), the batchsize, $B$, (middle) and the number of samples, $M$, (right) change for different values denoted by (+, □, ○, △). At each of these points (marks), we perform an experiment as illustrated in Figure 1, find $\mathcal{L}(\theta_{t_{95}})$ to compute the AE, and record the time corresponding to $t_{95}$. The results are averaged over five repetitions.

evaluation of performance of learning algorithms [43]. Traditional techniques usually define overlap (or separation) of components only in terms of their mean vectors and maximum eigenvalues of the covariance matrices, not accounting for their rotation and mixing weights (see [36] for a detailed treatment of the problem). We therefore use a more objective measure of the clustering complexity defined by the total probability of misclassification [41], which allows to generate data with a user-defined degree of maximum pairwise overlap, $\omega$.

*Experiment settings:* We generate the parameters of (1), and the corresponding dataset, uniquely for a given quadruple $(D, K, N, \omega)$. Therefore, the parameters of the generative model are known and we can measure and display the convergence of the training log-likelihood, $\mathcal{L}(\theta_t)$, compared to the exact log-likelihood, $\mathcal{L}(\theta)$, for $t = (1, \ldots, T)$. We are further interested in the absolute error between the training log-likelihood at the iteration of reaching 95% of its maximum value, $t_{95}$, and the exact log-likelihood, i.e. $AE = |\mathcal{L}(\theta_{t_{95}}) - \mathcal{L}(\theta)|$.

We also measure the computational time until reaching $t_{95}$. We have used 95% of the maximum value instead of the maximum value to prevent cases, where the model oscillate around target value, making the estimate of convergence time very noisy (for example MCSAEM in Figure 1).

*Algorithms:* The GMMs belong to the exponential family of probability distributions. This allows us to find a closed-form, recursive, solution of (6), relying on a Robbins-Monro type of the step-size sequence, $(\gamma_t)_{t=1}^T$, [7, 44]. In this setting, we compare our MHSAEM algorithm with a number of related methods in Table 1. Note we use the acronyms U and TF to specify the proposal distribution of the MHSAEM algorithm (Section 4.3). However, we do not use the O-proposal, since the MHSAEM-O algorithm is equivalent to the MCSAEM algorithm. All the SA-variants in Table 1 use a minibatch of size $B$. The key quantity to reduce the number of evaluated components and/or sufficient statistics in the SSAEM, TSAEM, MCSAEM and MHSAEM algorithms is collectively denoted by $M$ (Section 5). Note that we always keep $M = \bar{M}$ in the TSAEM algorithm (see Figure 1 and 2 for concrete numbers). We use the step-size given by $\gamma_t = 1$ for $t = 1, \ldots, 50$ and $\gamma_t = 0.05$ otherwise. In this section, to counteract the issue of attaining poor local optima, we equip *all* algorithms with the anti-annealing schedule $(\beta_t)_{t=1}^T$, starting with $\beta_1 = 0.1$, reaching $\beta_{2/3T} = 1.2$, and decreasing back to $\beta_T = 1.0$, see [43] for details. The initial estimates of: (i) $\mu_k$ are uniformly drawn from the unit hyper-cube, (ii) $\Sigma_k$ are fixed to unit diagonal matrix, and (iii) $\pi_k$ are uniformly drawn from the unit interval (followed by normalization).

*Results:* Figure 1 shows that the EM [13] and SAEM [44] algorithms take the longest time to converge, attaining a poor local optima. On the other hand, the MCSAEM [1] and MHSAEM (U and TF) algorithms achieve $\mathcal{L}(\theta_{t_{95}})$ closest to the likelihood $\mathcal{L}(\theta)$ of the true model. Moreover, both MHSAEM algorithms reach this value in the shortest time compared to all the other methods. The SSAEM [24] and TSAEM [18] algorithms are comparable in terms of the computational time, but they both provide the lowest $\mathcal{L}(\theta_{t_{95}})$. In Figure 2, we investigate sensitivity of fitting the model to

increasing values of $K$, $B$ and $M$ by measuring the time and the likelihood again. In all the cases, the proposed MHSAEM algorithms achieve the lowest AE in the shortest time.

SSAEM and TSAEM algorithms failed to converge for $M > 2$ and for $K > 50$ respectively. We believe this is caused by selecting only $M$ maximal probabilities in the SSAEM (or distances in the TSAEM) algorithm (Section 5), which prevents certain, but not a negligible number of, components from being updated, thus providing only a crude approximation of $p_\theta(z|x)$. The results then suffer from substantial variational gap to the exact log-likelihood (Figure 1). On the contrary, MH sampler provides samples which consistently approximate $p_\theta(z|x)$ despite evaluating much lower number of components in each step.

## 6.2 Sum-product transform networks

The sum product networks (SPNs) are a deep learning extension of finite mixture models. They can be interpreted as a mixture of trees [60], where each tree corresponds to a component. Therefore, they can be cast into the form of (1), but the number of components grows exponentially with their depth. In this section, we use recently proposed SPTNs which introduce additional transformation nodes to provide better expressiveness than the SPNs (SPTNs effectively generalize SPNs and flow models into one large family of models).

*Experimental settings:* We use 19 real datasets from the UCI database [16, 37, 35, 54], preprocessed in the same way as in [46]. For each experiment, we randomly split the data into 64%, 16% and 20% for training, validation and testing, respectively. We calculate the average log-likelihood on the test set and measure again the time to reach 95% of the maximal training log-likelihood, $\mathcal{L}(\theta_{t_{95}})$.

To evaluate various (possibly shallow and/or deep) architectures of SPTNs, we fit each dataset with all the following combinations of hyper-parameters[1]: $s \in (8, 32, 128)$, $b \in (2, 4, 6, 8)$, $l \in (2, 3, 4)$, where $s$ is the number of children of each sum node, $b$ is the number of partitions of each product node, and $l$ is the number of layers (one layer contains sum and product nodes). The number of components of the SPTN, after its conversion into (1), is given as follows: $K = s^l$. Note that the maximum number of components for the investigated parameters of the SPTN is 268,435,456. To reduce the space of possible architectures, we restrict ourselves only to (i) the leaf nodes given by $\mathcal{N}(0, \mathbf{I})$; (ii) affine transformations fixed to the singular value decomposition, choosing the the Givens parameterization for the unitary matrices [47]; and (iii) no sharing of any type of nodes [47].

*Algorithms:* We evaluate only on the MHSAEM-U algorithm—due to its favourable computational complexity and simplicity—and compare it with the stochastic gradient-descent (SGD) algorithm, which is routinely used to train SP(T)Ns [45, 47]. In this case, SGD in each iteration performs computations over all subtrees of the network, whereas the MHSAEM-U algorithm computes with only $M = 1$ subtrees, thus we should observe speed-up of the computations. In our implementation, both these methods perform optimization of their respective objective functions—the log-likelihood (2) for SGD and the EM objective (8) for MHSAEM-U—via the use of the automatic differentiation and the ADAM optimizer [29], using $B = 100$ and $T = 20000$.

*Results:* Since each dataset might benefit from a different architecture, Table 6.2 shows the test log-likelihood of the architectures selected according to the best likelihood measured on the validation set and the corresponding speed-up. The test log-likelihoods reveal that the MHSAEM-U algorithm outperforms the SGD algorithm on 10 out of 19 datasets, which was not originally the goal, but the added stochasticity helps to escape poor local minima. The speed-up demonstrates lower computational complexity of the MHSAEM-U algorithm on 17 out of 19 datasets, which was the main goal. The `magic-telescope` and `wine` datasets show approximately $102\times$ and $75\times$ speed-up, respectively, while on very small datasets (`pima-indians` and `iris`), the SGD is faster due to effective implementation. In the supplementary material, we present Table 3, exhibiting the same trends on a fixed architecture.

## 6.3 Mixtures of real NVP flows

We consider another class of mixture models (1), where each component $p_{\eta_z}(x|z)$ is transformed by the flow model—real NVP [15]. These transformations are parameterized via deep neural networks, allowing for flexible adjustment of the learning capacity of each component.

---

[1]We have set a hard limit to train a single model to 24h, which is default on our Slurm cluster.

Table 2: The speed-up and test log-likelihood, $\mathcal{L}^{\text{test}}$, for the SGD and MHSAEM-U algorithms. The test log-likelihood (higher is better) is computed for the best model, with the corresponding $K$, which is selected based on the validation log-likelihood. The speed-up is computed as the ratio of MHSAEM-U to SGD, i.e. their time to reach 95% of the training log-likelihood. The results are averaged over five repetitions. Then, the higher test log-likelihood is highlighted with bold blue, and and no speed-up is highlighted with red. The average rank is computed as the standard competition ("1224") ranking [14] on each dataset (lower is better).

| | Sum-product transform networks | | | | | Mixtures of real NVP flows | | | | |
| | SGD | | | MHSAEM-U | | | SGD | | | MHSAEM-U | |
| dataset | speed-up | $\mathcal{L}^{\text{test}}$ | $K$ | $\mathcal{L}^{\text{test}}$ | $K$ | speed-up | $\mathcal{L}^{\text{test}}$ | $K$ | $\mathcal{L}^{\text{test}}$ | $K$ |
|---|---|---|---|---|---|---|---|---|---|---|
| breast-cancer-wisconsin | 4.66 | -4.66 | 64 | **1.43** | 1024 | **0.63** | -99.85 | 32 | **-39.31** | 128 |
| cardiotocography | 10.55 | **59.52** | 512 | 31.04 | 1024 | 9.85 | 54.34 | 32 | **56.08** | 128 |
| magic-telescope | 102.53 | **-3.65** | 512 | -5.03 | 1024 | 3.74 | **-3.97** | 8 | -4.22 | 8 |
| pendigits | 4.89 | **0.88** | 1024 | -4.86 | 16384 | 4.17 | **1.46** | 8 | 0.48 | 8 |
| pima-indians | **0.37** | -8.54 | 64 | **-7.62** | 64 | 1.35 | -20.09 | 128 | **-16.33** | 128 |
| wall-following-robot | 3.43 | **1.84** | 1024 | -11.3 | 16384 | 22.21 | **-14.26** | 128 | -17.56 | 128 |
| waveform-1 | 4.35 | -26.14 | 64 | **-23.91** | 1024 | 3.72 | -34.12 | 8 | **-33.42** | 8 |
| waveform-2 | 4.82 | -26.21 | 64 | **-23.91** | 1024 | 4.12 | -34.15 | 8 | **-33.64** | 8 |
| yeast | 20.57 | **10.26** | 512 | 5.18 | 1024 | 14.49 | 6.61 | 128 | **9.59** | 128 |
| ecoli | 1.86 | -5.5 | 64 | **-0.22** | 1024 | 2.15 | -11.37 | 128 | **-10.64** | 128 |
| ionosphere | 1.88 | -20.27 | 64 | **-5.93** | 512 | 2.74 | -87.01 | 128 | **-42.75** | 128 |
| iris | **0.23** | -10.65 | 64 | **-1.49** | 16384 | 3.28 | -16.34 | 128 | **-9.21** | 32 |
| page-blocks | 12.18 | **12.21** | 512 | 6.84 | 1024 | 44.95 | 17.13 | 128 | **17.94** | 32 |
| parkinsons | 1.46 | -21.85 | 64 | **0.5** | 512 | 3.09 | -566.58 | 128 | **-33.31** | 32 |
| sonar | 2.96 | -95.39 | 512 | **-69.29** | 512 | 2.52 | -622.2 | 128 | **-88.81** | 128 |
| statlog-segment | 1.44 | **47.35** | 512 | 26.53 | 16384 | 38.49 | 35.84 | 128 | **42.04** | 32 |
| statlog-vehicle | 2.97 | **-4.25** | 64 | -5.45 | 1024 | 6.78 | -31.34 | 32 | **-26.43** | 128 |
| wine | 75.42 | -25.99 | 1024 | **-13.27** | 1024 | 2.05 | -171.58 | 128 | **-25.57** | 128 |
| rank | | 1.56 | | **1.44** | | | 1.83 | | **1.17** | |

*Experimental settings:* We use the same experimental settings and evaluation metrics as in Section 6.2. We apply the mixture model on all datasets, changing the number of components as follows: $K \in (8, 32, 128)$. Each real NVP-based component in the mixture model has (i) the translation function parameterized via multi-layer perceptron with a single hidden layer of dimension 10, using the rectified linear activation function; and (ii) the scale function parameterized via the same network except with the hyperbolic tangent activation function. We do not use the batch normalization [15] and we stack two layers of the translation-scale transformation (we have used implementation from [20]).

*Algorithms:* The algorithms and their settings are the same as those in Section 6.2.

*Results:* The experimental results are presented in right part of Table 6.2. They are similar to those obtained in the previous section. In terms of the test log-likelihood, the MHSAEM-U algorithm outperforms the SGD algorithm on all but three datasets, and it provides a substantial speed-up on all datasets except one. The test likelihood of models with the real NVP flows is most of the time worse than that of SPTNs with the affine transformations. As explained in the supplementary, this is due to the overfitting, which has been observed in [47].

# 7   Conclusion

This paper has presented a method to decrease computational complexity of fitting mixture models, including their generalizations, such as sum-product-(transform) networks and mixtures of flow models. The speed-up is achieved by evaluating and updating only a single component (per iteration), where the Metropolis-Hasting algorithm ensures sampling of components from a proper posterior. An experimental comparison on all three classes of models mentioned above confirmed the theoretical expectations. The method significantly speeds-up the fitting time and, importantly, without sacrificing the quality of the fit. In fact, the likelihood was better than that of the models fitted by the EM algorithm or the SGD algorithm in more than 50% of cases. We attribute this to higher stochasticity, which helps to escape from poor local minima.

In the experiments, the proposed method has used a uniform proposal distribution in the MH sampler. Despite outperforming the alternative methods, we conjecture that this limits the speed of convergence. Therefore, we believe that there is still a room for improvement in the implementation. We plan to address these issues in future work.

## 8 Broader impact statement

The presented method decreases the computational complexity of fitting large (and deep) mixture models, which leads to five to hundred time speed-up depending on a size of the problem (although negative exceptions occurs). We believe this line of research, which we want to continue, to have important benefits. First, it is directly related to decrease in energy consumption and in production of $CO_2$ (we expect similar rates as the speedup). Second, it has a positive effect on financial aspects of deploying (and experimenting with) mixture models. Third, it decreases the hardware requirements, as in all experiments presented above the model was fitted on a single-core.

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
