# A supplementary material - Fitting large mixture models using stochastic component selection

## Abstract

1      This is a supplementary material which provides additional details and experiments
2      for the paper entitled: *"Fitting large mixture models using stochastic component*
3      *selection"*[1].

## 9    Gaussian mixture models

5   In this section, we provide more experiments with the GMMs presented in Section 6.1.

### 9.1    Counteracting weak local optima with deterministic annealing

7 The EM algorithm converges to a local optimum in a finite number of iterations [6]. In simple
8 scenarios—where the number of components, $K$, is low—making a few attempts with different
9 initial conditions may successfully lead to finding the global optimum. However, for large $K$, the
10 likelihood surface is complicated and contains several weak local optima. To counteract this issue,
11 we apply the deterministic annealing [5], which—similarly to the simulated annealing [2]—has
12 its roots in thermodynamics and the maximum entropy principle. Indeed, maximizing the entropy
13 term, $\mathcal{H}(\hat{\theta})$, in (3) w.r.t. a variational posterior distribution, $q_{\hat{\theta}}(z|x)$, leads to $q_{\hat{\theta}}(z|x) \propto p_{\hat{\theta}}(z,x)^{\beta}$,
14 where $\beta$ is the inverse temperature parameter. This does not change the form of the EM objective,
15 $\mathcal{Q}(\theta)$, expect that its original posterior, $p_{\hat{\theta}}(z|x)$, is replaced by $q_{\hat{\theta}}(z|x)$. Then, it can be shown that
16 $\beta$ modifies the log-likelihood, $\mathcal{L}(\theta)$, [5]. Specifically, for $\beta \to 0$ (high temperature), the likelihood
17 surface is nearly uniform, having a single global optimum, whereas for $\beta \to 1$ (low temperature),
18 $q_{\hat{\theta}}(z|x) \to p_{\hat{\theta}}(z|x)$, having several local optima. Therefore, the key requirement is to change $\beta$ via a
19 pre-specified annealing schedule, $(\beta_t)_{t=1}^{T}$, such that the global optimum slowly appears and is thus
20 easier to find. This principle is applicable to all the algorithms in Table 1.

21 We resort to the deterministic anti-annealing [4], which improves the convergence speed over the
22 deterministic annealing by admitting $\beta > 1$. Specifically, it forms $(\beta_t)_{t=1}^{T}$ as follows: start with
23 $\beta_1 = \beta_{\min}$, then increase to $\beta_{\tau} = \beta_{\max} > 1$, where $\tau < T$, and finally decrease back to $\beta_T = 1$.

24 Figure 3 compares the various EM algorithms in Table 1 with (right) and without (left) the anti-
25 annealing schedule. We observe that the EM, SAEM and MCSAEM algorithms with the annealing
26 find a better local optimum compared to the corresponding counterparts without the annealing.
27 However, the annealing seems to have no effect on the poor optimum attained by the SSAEM and
28 TSAEM algorithms. Similarly, there is only a slight improvement in the optimum reached by the
29 MHSAEM algorithm (both U and TF proposals). Still, this is the best performance compared to all
30 the other algorithms, having the estimated log-likelihood very close to the exact one. We attribute the

---

[1]The code to reproduce all experiments in the main paper and the supplementary material is available at: `https://drive.google.com/drive/folders/1foHlyCCHS8N2Odw6sXQI9bDCRTQIuwkU?usp=sharing`.

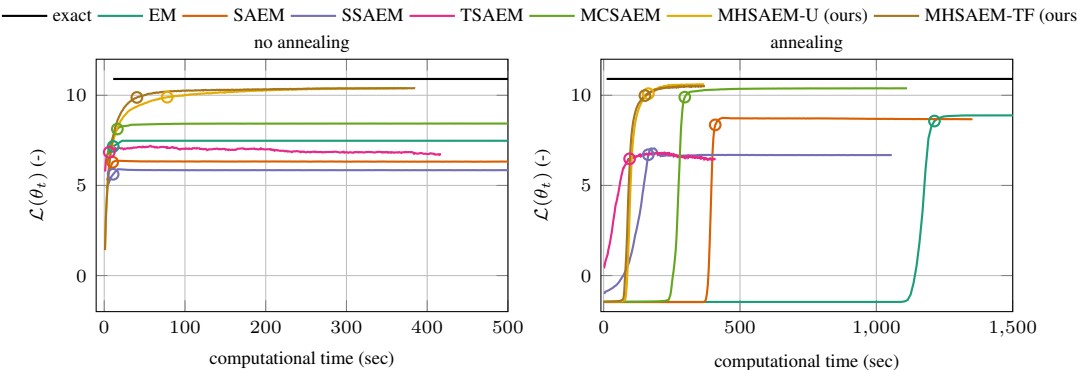

Figure 3: The training log-likelihood, $\mathcal{L}(\theta_t)$, versus the computational time (in seconds). The algorithms either do not use (left) or use (right) the anti-annealing schedule described in Section 6.1. Here, on the x-axis, the computational time at a current iteration, $t$, is obtained by accumulating the time from the previous iterations. $\bigcirc$ corresponds to $\mathcal{L}(\theta_{t_{95}})$, where $t_{95}$ is the iteration of reaching 95% of $\max \mathcal{L}(\theta_t)$. The projection of $\bigcirc$ on the x-axis gives the time to reach $\mathcal{L}(\theta_{t_{95}})$. This experiment was performed with the following settings: $(D, K, N, \omega, B, M, T) = (10, 100, 10k, 0.1, 200, 2, 20k)$. The results are averaged over five repetitions.

small difference to the overlap of a small portion of less representative clusters, preventing them from being represented via their sufficient statistics to an adequate degree.

Overall, albeit the annealing enhances the algorithms' ability to find a better optimum, it is rather the stochastic nature of the sampling-based algorithms what provides a better fit of the model.

## 9.2 The computational complexity under various operating conditions

Recall that the computational complexity of our MHSAEM algorithm is $\mathcal{O}(BMDT)$ (Section 4), where $B$ is the batchsize, $M$ is the number of samples, $D$ is the dimension of data and $T$ is the total amount of iterations. Although there is no direct dependence on $K$, there are certain aspects of the algorithm that still do depend on $K$, e.g. all operations associated with the memory management of selecting $M$ of $K$ sufficient statistics. Therefore, we investigate how the computational complexity and the training performance depend on $K$ while changing $D$, $M$ and $B$. We leave $T$, since this affects all algorithms in the same way. We use the performance criteria from Section 6.1.

*Batchsize.* All the SA-algorithms in Table 1 sub-sample data with a minibatch of size $B$. Each algorithm processes each datapoint in the minibatch w.r.t. a different number of components in a different way. Figure 4 shows the AE versus the computational time for various $K$ and $B$. Note we include the EM algorithm, which does not depend on $B$, just for a comparison. From all the $B$-dependent variants, the SAEM algorithm has the highest computational complexity, since it processes all $K$ components for each datapoint in the minibatch. The MHSAEM algorithms attain the best AE in the shortest time, assessing only $M$ components for each datapoint in the minibatch. Although the MCSAEM algorithm provides a similar AE, its computational time grows faster with increasing $B$. This is due to that the posterior distribution has to be evaluated for all $K$ components before the MC sampling. The SSAEM algorithm preserves a similar computational complexity as the MHSAEM algorithms for all $B$. However, it delivers a poor AE. The TSAEM algorithm fails to converge in this experiment.

*Number of samples.* The number of samples (the MCSAEM and MHSAEM algorithms), or selection points (the SSAEM and TSAEM algorithms), $M$, determines the amount of components processed for each datapoint in the minibatch. In other words, this quantity determines the number of sufficient statistics to be updated at each iteration of a given algorithm. Since this feature is common across all $M$-dependent algorithms in Table 1, we can expect that the computational times will differ approximately by a constant factor, which is due to different sampling, or selection, mechanisms. Indeed, this is seen in Figure 5, where there is approximately the same distance between the MCSAEM and MHSAEM algorithms for all $M$. This can also be observed for the SSAEM algorithm, except the last case with $M = 8$. We believe that this is caused by the application of the fast stochastic sorting

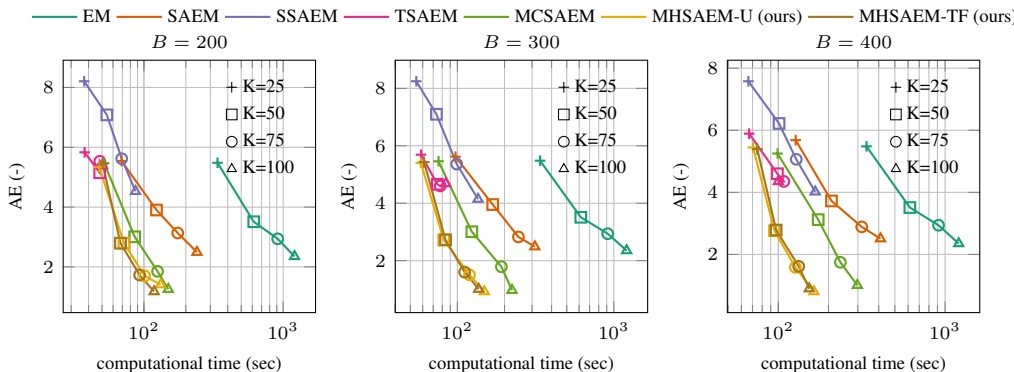

Figure 4: The absolute error, AE $= |\mathcal{L}(\theta_{t_{95}}) - \mathcal{L}(\theta)|$, versus the computational time (in seconds). All experiments use the following settings: $(D, K, N, \omega, B, M, T) = (2, K, 20k, 0.001, B, 8, 40k)$, where the number of components, $K$, changes for different values denoted by $(+, \square, \bigcirc, \triangle)$ and the batchsize is $B = 200$ (left), $B = 300$ (middle) and $B = 400$ (right). At each of these points (marks), we perform an experiment as illustrated in Figure 1 (right), find $\mathcal{L}(\theta_{t_{95}})$ to compute the AE, and record the time corresponding to $t_{95}$. The results are averaged over five repetitions.

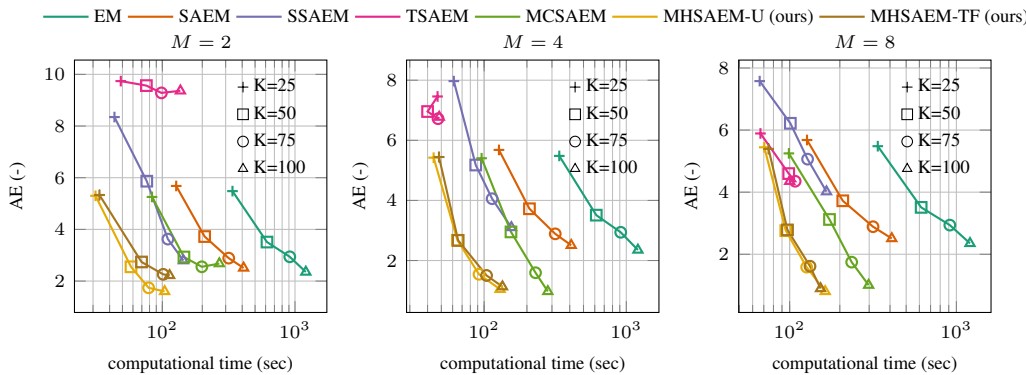

Figure 5: The absolute error, AE $= |\mathcal{L}(\theta_{t_{95}}) - \mathcal{L}(\theta)|$, versus the computational time (in seconds). All experiments use the following settings: $(D, K, N, \omega, B, M, T) = (2, K, 20k, 0.001, 400, M, 40k)$, where the number of components, $K$, changes for different values denoted by $(+, \square, \bigcirc, \triangle)$ and the number of samples is $M = 2$ (left), $M = 4$ (middle) and $M = 8$ (right). At each of these points (marks), we perform an experiment as illustrated in Figure 1 (right), find $\mathcal{L}(\theta_{t_{95}})$ to compute the AE, and record the time corresponding to $t_{95}$. The results are averaged over five repetitions.

algorithm [3], which increases the computational time due to different initial state of the unsorted vector of the posterior probabilities at each iteration, $t$. Again, the TSAEM algorithm delivers a rather unsystematic behaviour. Similarly as before, we include the $M$-independent EM and SAEM algorithms for a comparison.

*Dimension.* The dimension, $D$, has a direct impact on the computational complexity of updating the expected sufficient statistics, evaluating the coditional likelihood, $p_{\hat{\theta}}(x|z)$, and computing the parameter estimates, $\hat{\theta}$. All the algorithms in Table 1 differ in the way they perform these elementary operations. Therefore, in Figure 6, we compare the computational time for various choices of $D$. Unfortunately, the values, $D \in (2, 4, 8)$, are too low to demonstrate clear differences among the algorithms. This is caused by that the time to carry out the algebraic operations with such similar low dimensions is too short to stand out against the overhead associated with other computational steps in the algorithms. This is easy to deal with by repeating the experiment for, say, $D \in (1, 10, 100)$, or further optimizing our implementation. We will address this in future work. Nonetheless, this experiment shows consistent behaviour of our MHSAEM algorithm under varying $D$.

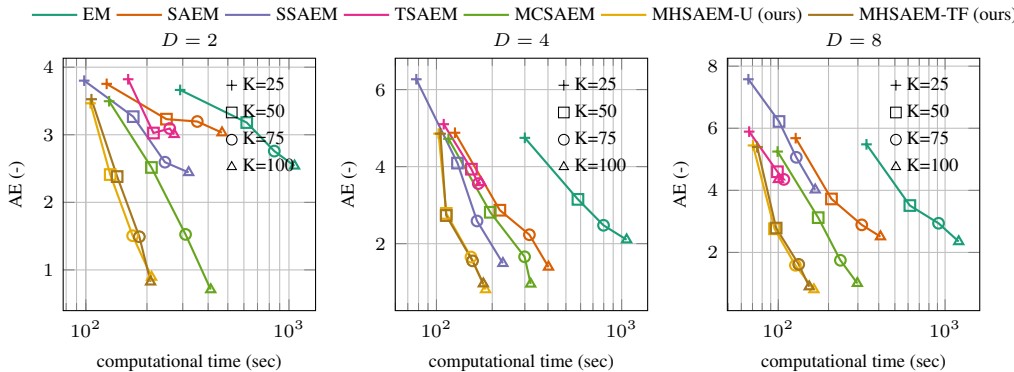

Figure 6: The absolute error, AE $= |\mathcal{L}(\theta_{t_{95}}) - \mathcal{L}(\theta)|$, versus the computational time (in seconds). All experiments use the following settings: $(D, K, N, \omega, B, M, T) = (D, K, 20k, 0.001, 400, 8, 40k)$, where the number of components, $K$, changes for different values denoted by $(+, \square, \bigcirc, \triangle)$ and the dimension of data is $D = 2$ (left), $D = 4$ (middle) and $D = 8$ (right). At each of these points (marks), we perform an experiment as illustrated in Figure 1 (right), find $\mathcal{L}(\theta_{t_{95}})$ to compute the AE, and record the time corresponding to $t_{95}$. The results are averaged over five repetitions.

### 9.3   Efficiency of the proposal distribution

The proposal distribution is the key factor influencing the performance of the MH sampler. Therefore, we compare the proposal distributions discussed in Section 4.3. To demonstrate the impact of introducing the forgetting factor in the TF proposal, we define another tabular proposal, here abbreviated with T, which is given by $q_{\boldsymbol{\alpha}_i}(z_i) := \mathcal{C}(\boldsymbol{\alpha}_i)$, where $\boldsymbol{\alpha}_i$ is computed via the maximum likelihood estimate, $\boldsymbol{\alpha}_{i,t} = \frac{1}{t} \sum_{\tau=1}^{t} \mathbf{e}_{z_{i,\tau}}$, see Section 4.3 for details. Indeed, this is similar to the TF proposal, expect it does not use the forgetting factor, $\gamma_t$, specified by the Robbins-Monro sequence, $(\gamma_t)_{t=1}^{T}$. In the following experiments, we use $\gamma_t = 1$ for $t = 1, \ldots, 500$ and $\gamma_t = 0.1$ otherwise. We do not use annealing in the following experiments. To evaluate the proposal distributions, we compute the average acceptance ratio (AAR) over all $i \in (1, \ldots, N)$ at each iteration, $t$, i.e. $\mathrm{AAR}_t := \frac{1}{N} \sum_{i=1}^{N} \alpha(z_{i,t-1}, z_{i,t})$.

Figure 7 shows that, as one may expect, the acceptance ratio of the O-proposal converges close to one, and the corresponding log-likelihood quickly converges near to the exact value. The acceptance ratio of the U-proposal converges towards $\frac{1}{K}$, as the uniform proposal has exactly $\frac{1}{K}$ probability to sample the most representative component for a given datapoint. However, this proposal is less efficient, as can be seen from slower convergence of the corresponding log-likelihood. The TF-proposal improves the performance over the U-proposal by having a higher acceptance ratio and quicker convergence of the log-likelihood. However, the T-proposal fails to learn a sufficiently close representation of the exact posterior distribution $p_\theta(z|x)$, as seen from high acceptance ratio and poor log-likelihood. This result demonstrates that introducing the Robbins-Monro schedule is crucial for improving the performance of the T-proposal.

The experiments presented in Section 6.1, Section 9.1 and Section 9.2 show that the U-proposal and the TF-proposal have similar performance. This indicates that the TF-proposal looses its efficiency for high $K$. We plan to address this in future work. Nonetheless, from all these results, we observe that even the simple U-proposal achieves a substantial speed-up compared to the other algorithms in Table 1. Designing a better proposal distribution will further decrease the computational time of our MHSAEM algorithm.

## 10   Sum-product-transform networks and mixtures of real NVP flows

This section extends experiments with the SPTNs and the mixtures of real NVP flows given in Section 6.2 and Section 6.3, respectively.

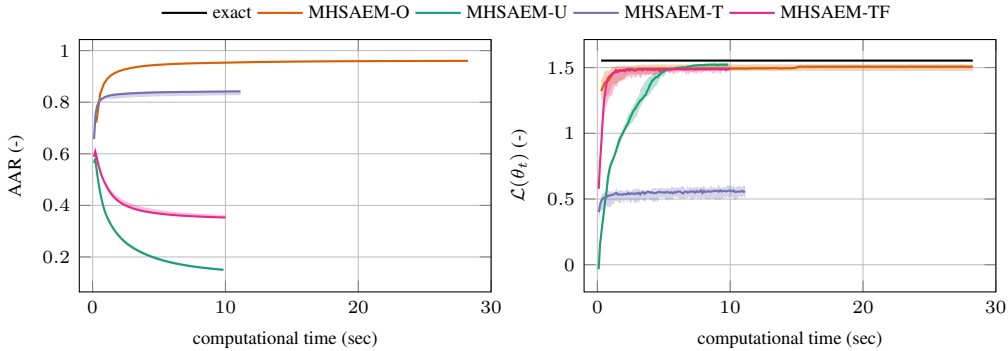

Figure 7: The average acceptance ration, $\mathrm{AAR}_t := \frac{1}{N} \sum_{i=1}^{N} \alpha(z_{i,t-1}, z_{i,t})$, (left) and the training log-likelihood, $\mathcal{L}(\theta_t)$, (right) versus the computational time (in seconds). This experiment was performed with the following settings: $(D, K, N, \omega, B, M, T) = (2, 10, 10k, 0.1, 200, 1, 20k)$. The results are averaged over ten repetitions. The solid line is the median and the shaded area is the inter-quartile range.

### 10.1 Computational complexity for a fixed architecture

In Table 2, we have compared the computational times and the test log-likelihoods for the best architectures (SPTNs) and the best models (mixtures of real NVP flows) selected based on the validation log-likelihood. This experiment allows to find the algorithm that trains the most suitable model in the shortest time. One the one hand, it reveals clear cases where the MHSAEM algorithm outperforms the SGD algorithm in terms of the test log-likelihood and the speed-up (e.g. `waveform`). In some situations this holds for the same architecture selected by both the SGD and MHSAEM algorithms (e.g. `wine`). We also see that using the SGD algorithm to fit an SPTN with $K = 1024$ takes longer time than using our MHSAEM algorithm to fit an SPTN with $K = 16384$, which is, however, not beneficial due to the lower values of the test log-likelihood (e.g. `pendigits`). On the other hand, there are situations showing no speed-up on small architectures, despite having a better log-likelihood (e.g. `pima-indians`). Here, the implementation of the SGD algorithm simply beast the one of the HMSAEM algorithm.

In those lines where $K$ is substantially different for the SGD and MHSAEM algorithms, it may be difficult to recognize the maximal achievable speed-up of the MHSAEM algorithm. Therefore, we compare these algorithms also for a fixed architecture (and model) in Table 3. For all datasets, our MHSAEM algorithm delivers a speed-up ranging approximately from 11 to 186 for SPTNs and from 1.3 to 38.6 for the mixtures of real NVP flows.

### 10.2 Overfit of the mixtures of real NVP flows

Certain models can be too complex for small and simple datasets. To consider this in our context, we present Table 4, which extends Table 2 by additionally presenting the training and validation log-likelihoods. Here, we observe that some instances of the mixtures of real NVP flows deliver too high training log-likelihoods. If we were select these models based on their validation log-likelihoods, the resulting test log-likelihood would be rather poor. This indicates a clear overfit. The results achieved by the SPTNs are more consistent in this respect, being less prone to the model overfit. Note, too, that the mixtures of real NVP flows and the SPTNs deliver comparable results in some cases.

Table 3: The speed-up and test log-likelihood, $\mathcal{L}^{\text{test}}$, for the SGD and MHSAEM-U algorithms. The test log-likelihood (higher is better) is computed for the SPTN with $K = 1024$ and the mixture of real NVP flows with $K = 128$. The speed-up is computed as the ratio of MHSAEM-U to SGD, i.e. their time to reach 95% of the training log-likelihood. The results are averaged over five repetitions. Then, the higher test log-likelihood is highlighted with bold blue. The average rank is computed as the standard competition ("1224") ranking [1] on each dataset (lower is better).

| | Sum-product transform networks | | | Mixtures of real NVP flows | | |
| | | SGD | MHSAEM-U | | SGD | MHSAEM-U |
| dataset | speed-up | $\mathcal{L}^{\text{test}}$ | $\mathcal{L}^{\text{test}}$ | speed-up | $\mathcal{L}^{\text{test}}$ | $\mathcal{L}^{\text{test}}$ |
|---|---|---|---|---|---|---|
| breast-cancer-wisconsin | 16.72 | -24.01 | **0.2** | 2.62 | -112.63 | **-39.31** |
| cardiotocography | 17.93 | **45.81** | 31.04 | 27.45 | 45.99 | **56.08** |
| magic-telescope | 24.21 | **-4.41** | -5.21 | 20.41 | -6.01 | **-4.96** |
| pendigits | 15.5 | **-0.58** | -5.96 | 17.82 | **-1.78** | -3.7 |
| pima-indians | 17.69 | -20.18 | **-8.3** | 1.35 | -20.09 | **-16.33** |
| wall-following-robot | 16.96 | **-6.05** | -16.98 | 22.21 | **-14.26** | -17.56 |
| waveform-1 | 101.37 | -39.16 | **-23.94** | 19.09 | **-39.06** | -40.41 |
| waveform-2 | 92.73 | -38.87 | **-23.95** | 20.32 | **-39.42** | -40.79 |
| yeast | 12.41 | 4.29 | **5.18** | 14.49 | 6.61 | **9.59** |
| ecoli | 29.45 | -14.02 | **-0.22** | 2.15 | -11.37 | **-10.64** |
| ionosphere | 13.78 | -30.25 | **-7.35** | 2.74 | -87.01 | **-42.75** |
| iris | 52.62 | -12.5 | **-1.76** | 1.45 | -16.34 | **-9.45** |
| page-blocks | 16.68 | **11.98** | 6.84 | 26.04 | 17.13 | **17.27** |
| parkinsons | 16.64 | -35.75 | **-1.15** | 3.22 | -566.58 | **-33.0** |
| sonar | 11.5 | -98.43 | **-88.33** | 2.52 | -622.2 | **-88.81** |
| statlog-satimage | 14.52 | 3.33 | **3.74** | 25.31 | **-7.38** | -17.89 |
| statlog-segment | 16.35 | **40.81** | 25.73 | 38.68 | 35.84 | **36.8** |
| statlog-vehicle | 23.43 | -22.52 | **-5.45** | 33.71 | -31.21 | **-26.43** |
| wine | 186.25 | -26.53 | **-13.5** | 2.05 | -171.58 | **-25.57** |
| rank | | 1.68 | **1.32** | | 1.74 | **1.26** |

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

Table 4: 95% of the maximal training log-likelihood, $\mathcal{L}(\theta_{t_{95}})$, validation log-likelihood, $\mathcal{L}^{\text{val}}$, and test log-likelihood, $\mathcal{L}^{\text{test}}$, for the SGD and MHSAEM-U algorithms (higher is better). The lines in this table correspond to those in Table 2, except we add the training and validation log-likelihoods and do not repeat the speed-up and the number of components, $K$ (see Table 2 for completeness). The results are averaged over five repetitions. Then, the higher log-likelihood is highlighted with bold blue. The average rank is computed as the standard competition ("1224") ranking [1] on each dataset (lower is better).

| | Sum-product transform networks | | | | | | Mixtures of real NVP flows | | | | | |
| | $\mathcal{L}(\theta_{t_{95}})$ | | $\mathcal{L}^{\text{test}}$ | | $\mathcal{L}^{\text{val}}$ | | $\mathcal{L}(\theta_{t_{95}})$ | | $\mathcal{L}^{\text{test}}$ | | $\mathcal{L}^{\text{val}}$ | |
| dataset | SGD | MHSAEM-U | SGD | MHSAEM-U | SGD | MHSAEM-U | SGD | MHSAEM-U | SGD | MHSAEM-U | SGD | MHSAEM-U |
|---|---|---|---|---|---|---|---|---|---|---|---|---|
| breast-cancer-wisconsin | **11.88** | 5.47 | -4.66 | **1.43** | -5.89 | **0.33** | 21.19 | **46.97** | -99.85 | **-39.31** | -70.33 | **-39.21** |
| cardiotocography | **59.72** | 30.36 | **59.52** | 31.04 | **59.09** | 30.06 | 58.21 | **62.95** | 54.34 | **56.08** | 52.7 | **54.35** |
| magic-telescope | **-3.08** | -5.24 | **-3.65** | -5.03 | **-3.75** | -5.11 | **-3.79** | -4.07 | **-3.97** | -4.22 | **-4.1** | -4.31 |
| pendigits | **4.89** | -4.21 | **0.88** | -4.86 | **0.77** | -4.77 | **5.09** | 3.94 | **1.46** | 0.48 | **1.66** | 0.77 |
| pima-indians | -7.22 | -7.4 | -8.54 | **-7.62** | -8.63 | **-7.54** | 1.97 | **11.52** | -20.09 | **-16.33** | -21.48 | **-16.61** |
| wall-following-robot | **15.9** | -6.75 | **1.84** | -11.3 | **-0.16** | -11.36 | **26.53** | 22.95 | **-14.26** | -17.56 | **-15.99** | -19.11 |
| waveform-1 | **-22.34** | -24.4 | -26.14 | **-23.91** | -25.93 | **-23.86** | **-20.94** | -21.01 | -34.12 | **-33.42** | -34.31 | **-33.6** |
| waveform-2 | **-22.17** | -24.4 | -26.21 | **-23.91** | -25.87 | **-23.86** | **-20.87** | -21.03 | -34.15 | **-33.64** | -34.02 | **-33.54** |
| yeast | **13.63** | 5.59 | **10.26** | 5.18 | **10.81** | 5.79 | 10.26 | **14.0** | 6.61 | **9.59** | 8.45 | **10.79** |
| ecoli | **6.3** | 0.65 | -5.5 | **-0.22** | -6.36 | **-0.83** | 5.55 | **10.78** | -11.37 | **-10.64** | -46.87 | **-11.66** |
| ionosphere | **33.58** | 6.87 | -20.27 | **-5.93** | -12.39 | **-1.87** | 43.3 | **74.58** | -87.01 | **-42.75** | -102.24 | **-41.88** |
| iris | **4.7** | -1.58 | -10.65 | **-1.49** | -12.11 | **-1.84** | 1.17 | **7.64** | -16.34 | **-9.21** | -13.88 | **-9.05** |
| page-blocks | **12.76** | 6.73 | **12.21** | 6.84 | **12.36** | 6.93 | 17.67 | **18.62** | 17.13 | **17.94** | 17.42 | **18.01** |
| parkinsons | **27.23** | 5.3 | -21.85 | **0.5** | -23.61 | **-0.25** | 33.97 | **50.33** | -566.58 | **-33.31** | -552.33 | **-33.46** |
| sonar | **158.64** | -16.24 | -95.39 | **-69.29** | -96.02 | **-72.51** | 78.05 | **142.83** | -622.2 | **-88.81** | -371.82 | **-87.63** |
| statlog-segment | **51.73** | 28.01 | **47.35** | 26.53 | **44.79** | 26.3 | 47.46 | **50.17** | 35.84 | **42.04** | 34.8 | **41.4** |
| statlog-vehicle | **3.43** | -4.0 | **-4.25** | -5.45 | **-4.83** | -5.31 | 14.79 | **22.51** | -31.34 | **-26.43** | -30.23 | **-25.92** |
| wine | **43.01** | -12.39 | -25.99 | **-13.27** | -26.17 | **-12.95** | 19.21 | **34.77** | -171.58 | **-25.57** | -262.69 | **-25.03** |
| rank | **1.0** | 2.0 | 1.56 | **1.44** | 1.56 | **1.44** | 1.72 | **1.28** | 1.83 | **1.17** | 1.83 | **1.17** |