# OpenReview forum: "Fitting large mixture models using stochastic component selection"
_NeurIPS.cc/2021/Conference — NeurIPS 2021 Submitted_

### Official Review · Reviewer_XENJ · 2021-07-14

**Rating:** 3
**Confidence:** 4

**Summary:**

The authors propose a sampling-based method to improve the efficiency of the E-M algorithm on mixture models. The method relies on performing the computations on a reduced number of MCMC-sampled components, rather than all of them.

**Limitations And Societal Impact:**

Yes.

**Main Review:**

The paper is well-structured, and I did not have a problem following the exposition.

The main idea of the paper is straightforward: instead of performing computations over all mixture model components, perform an approximate computation on a smaller number of components sampled via MCMC to reduce the total complexity of E-M. While reasonable, this idea by itself does not constitute a significant contribution, and there is no further theoretical analysis or discussion. The "TF" proposal distribution in section 4.3 could potentially be a novel contribution, but it is only discussed very briefly (see also following comments).

The experimental section, and specifically the two non-synthetic data experiments of sections 6.2 and 6.3, seem to have some major flaws:
- The authors do not even evaluate the previously discussed "TF" proposal in these experiments.
- The experiments seem to have been run with $M = 1$, which means just choosing a single component at random. Whether this choice is uniform or not depends on whether there is a burn-in period or not, which is also not discussed in the paper. I do not see the point in discussing an MCMC method if only a single sample is used in the end.
- I do not understand why there are such big differences in the test likelihood values in Table 2 between the two compared methods. In particular, why would the "SGD" method ever result in so much worse likelihoods? Is it some implicit regularization effect? Is the variance of the likelihood over different repetitions high? Why do the authors not show standard deviations, since their results are over five repetitions?

**Time Spent Reviewing:**

4

---

> ### Author Response · Authors · 2021-08-10
> **TF proposal distribution**
>
> *''The authors do not even evaluate the previously discussed "TF" proposal in these experiments.''*
>
> Figure 7 (the supplementary material) demonstrates the benefit of using the TF-proposal on a synthetic example with $K=10$. However, Figure 2 (the main paper) shows that there is a little difference between the U and TF proposals for $K=100$. This reveals that the TF proposal loses its efficiency for high $K$. For this reason and implementation complexity, we decided not to use the TF proposal in Sections 6.2 and 6.3. The U-proposal still delivers a solid speed-up over alternative methods (Section 6). Nevertheless, we admit that this simple proposal limits the convergence (Section 7) and that designing a better proposal distribution can increase the speed-up even further.
>
> *''The experiments seem to have been run with $M=1$, which means just choosing a single component at random. Whether this choice is uniform or not depends on whether there is a burn-in period or not, which is also not discussed in the paper. I do not see the point in discussing an MCMC method if only a single sample is used in the end.''*
>
> The chain is connected across the iterations, hence even for $M=1$, we produce a Markov chain of length $T$, i.e. $(z_{i,t})^T_{t=1}$. This has been stated in the fourth paragraph of Section 4.1: ''... at every current iteration $t$, we continue to extend the chain from the point where we left at the previous iteration, $t-1$, by taking ...'' (see also the sixth line of Algorithm 1). This property has the effect that the burn-in phase takes place at the beginning of the iterations, where the generated non-i.i.d. samples---which are inappropriate to estimate $\theta$---are forgotten via the Robbins-Monro step-size schedule (Section 4.2). After the burn-in phase, the samples are approximately i.i.d. and are used to estimate $\theta$.
>
> *''I do not understand why there are such big differences in the test likelihood values in Table 2 between the two compared methods. In particular, why would the ``SGD" method ever result in so much worse likelihoods?''*
>
> The SGD and MHSAEM algorithms are just local optimization methods, i.e. they converge only to a local optimum. We attribute the success of our MHSAEM algorithm to the stochastic sampling of the latent variable, a feature that is not involved in the SGD algorithm. This added stochasticity allows the MHSAEM algorithm to better explore the likelihood surface and thus increase the chance to find a better local optimum.
>
> *''Is it some implicit regularization effect?''*
>
> We do not think so. We expect that it is more related to the exploration/exploitation trade-off.
>
> *''Is the variance of the likelihood over different repetitions high? Why do the authors not show standard deviations, since their results are over five repetitions?''*
>
> We had the error bars in Figures 1 and 2, but we decided to not include them as they were small in the reported log-scale (as mentioned in the checklist). We did not include the standard deviations in Table 2 due to space limitations. We will add this to the paper.

---

> > ### Author Response · Authors · 2021-08-27
> > **Clarification**
> >
> > Please, see our reply to the following comment: https://openreview.net/forum?id=ki4eJ1fSJNq&noteId=9BIgwv3grCJ

---

### Official Review · Reviewer_iZc8 · 2021-07-16

**Rating:** 5
**Confidence:** 4

**Summary:**

This paper proposes a new method for scalably fitting mixture models. The authors tackle the problem of fitting with a large number of mixture components. While there exist some works handling this problem, these methods are all limited to mixtures of exponential family distributions, whereas the authors’ method applies to generic mixture models (as long as the model provides a likelihood p(data point i | cluster k)). They do so by proposing a modified version of the EM algorithm: instead of exactly computing the expectations in the E step, the authors propose to use a Metropolis Hastings algorithm to approximate the required expectations. In experiments, the authors show that their method (1) outperforms pre-existing large-K methods (where such previous applications are applicable), and (2) show that their method outperforms standard stochastic gradient descent in settings where other large-K methods are not applicable. The authors’ method seems to work well, but I have a few clarification questions about the technical ideas behind the proposed algorithm and its empirical evaluation. Currently, I put a borderline score for the paper, but I’m definitely open to increasing it after discussion.

**Limitations And Societal Impact:**

The authors do include a broader impact statement, but it's over the nine page limit. Note that the broader impact statement is supposed to be included within the page limit of the main text.

In their paper checklist, the authors write [N/A] for whether their datasets were obtained with consent and whether their datasets contain personally identifiable information or offensive content. Their datasets were taken from the UCI repository, so presumably neither of these things is true, but that doesn't mean it isn't relevant to this paper. I think a quick appendix about the data used would resolve this issue.

**Main Review:**

---------------------------------------------

**Update after discussion with authors and other reviewers**

Thanks to the authors for really engaging in the discussion period. I think it's become clear that some major revisions are needed in order to fully describe / experiment with the proposed E-step approximation. I've actually changed my score to a weak reject because of this. I do want to be clear that this is only because of the limits of the conference review cycle, where reviewers don't see an updated draft. In particular, I think a revision is likely to result in a strong accept, but the revisions are too substantial to accept without seeing them. Just to summarize, there are two things that I think remain:

1. The main issue is the need for a more targeted discussion and experimentation surrounding the proposed E-step approximation. I think the directions in terms of references that the authors have brought up will strengthen things here a lot. And the proposed experiment seems like a great step in the right direction on the empirics front. Personally, I think using a continuing Markov chain between optimization iterations is pretty interesting and surprising, so I really think the paper would be both more complete and more interesting with a deeper investigation of this idea.

2. This is a lot more minor, but I do want to argue a little bit more about the stopping rule used in the experiments. I agree that adding some tolerance $\delta$ will give an extra degree of freedom in the experiments. But, since this is how the method will be used in practice, it seems worth looking into. I don't mean that it's necessary to have a whole extra set of experiments investigating the sensitivity with respect to $\delta$. It's just good to make the experiments as close to reality as possible, and it doesn't seem like there's a strong reason to not make the experiments more realistic in this regard.



---------------------------------------------

I’ve listed my main comments/questions as (A), (B), (C) below.

(A) The use of Monte Carlo to help with an intractable E step is not a new idea, and I think the paper needs more discussion of the related work in this area. For example, see Levine and Casella (2001) (reference below). Ideas from this literature might be helpful for the current paper – e.g. Levine and Casella (2001) recommend updating the Monte Carlo samples from the previous E step via importance sampling, rather than re-running a MCMC method. Still, I don’t know that other work has handled the issue of a high-dimensional parameter space (here, the number of mixture components K is large), which sounds like an important addition over pre-existing work.

(B) I have a few technical questions about the presented algorithm.
1. The hope of the method is that we can effectively approximate the E step by drawing $M \ll K$ samples from the distribution $p(z_i | x_i)$. It seems that in some sense this is automatically impossible – the distribution $p(z_i | x_i)$ has support of size $K$, so we can’t, in general, approximate expectations over this distribution with only $M$ samples. So I think it’s unlikely that, in general, “the chain passes the transition period (the burn-in phase)” as the authors note. Presumably the authors’ method either relies on some regularity conditions (maybe the distribution $p(z_i | x_i)$ is mostly concentrated on a few points?) or maybe performing the M-step with an only approximate objective is fine, or maybe something else is happening. In any case, can the authors clarify with discussion and/or additional experiments what is driving the success of their method?
2. On lines 130-138, I think the authors are proposing to somehow reuse the samples from $p(z_i | x_i)$ between iterations. But the distribution being sampled from is changing from iteration to iteration (i.e. it depends on the parameters $\theta$ estimated from the M-step). What do the authors mean on line 131 when they talk about samples being “recycled over the iterations”?
3. The authors parameterize the mixing weights $\pi_k$ by a $K$ real-valued parameters $\nu_1, \dots, \nu_K$, and then set $\pi_k = e^{\nu_k} / \sum_{l=1}^K e^{\nu_l}$. But this transformation leaves a degree of freedom unconstrained (i.e. scaling all $\nu_k$ by the same constant $c$ will leave the $\pi$’s unchanged), meaning there is no unique optimum of the log-likelihood. This seems potentially undesirable, but maybe isn’t a huge deal.

(C) I also have a few questions about the empirical evaluation of the authors’ method.
1. For the purposes of considering an algorithm terminated, the authors stop each algorithm when it reaches 95% of the maximum likelihood value that it will achieve. I have two questions about this. The first is why is this a reasonable metric? Can we be sure that the fitted parameters will not change much as we traverse the last 5% of the loss? Second, this doesn’t necessarily seem like a good way to assess convergence of practical algorithms. In particular, the only way to define this stopping criteria is to first run the algorithm to termination so that we know what the final log likelihood value is, and then go back and stop it 95% of the way there. Can the authors argue that this does not bias the results at all? For example, it could be that it takes much longer for their algorithm to traverse the final 5% of the likelihood than other algorithms.
2. The authors choose a particular annealing schedule for use across all algorithms. How did they choose this schedule? Does it definitely not bias the results in favor of one algorithm or another?
3. Are all the experiments done with multiple random restarts for each method? Figure 1 says that “the results are averaged over five repetitions”, but it’s not clear to me how these repetitions are done. Are these five different random initializations? Or five different runs from the same initialization? Each curve in Figure 1 looks like the run from a single random seed, so it seems to be the latter. I think there should be multiple random seeds used so that we know that, for example, TSAEM (the fastest algorithm to reach its optimum in Figure 1) isn’t just getting unluckily stuck in a local optimum in Figure 1.
4. Why isn’t MHSAEM-TF used in the experiment in Section 6.2? The authors state that it is due to the “favorable computational complexity and simplicity” of MHSAEM-U, but it seems like in Figure 2 the U and TF versions of these algorithms have just about the same performance.
5. Similarly to point #3, are the results in Table 2 run over multiple random restarts? If so, there should be standard errors reported so that we know how significant the differences are.

-----Citations--------

Richard A. Levine and George Casella. Implementations of the Monte Carlo EM Algorithm. Journal of Computational and Graphical Statistics , Sep., 2001, Vol. 10, No. 3


**Time Spent Reviewing:**

5

---

> ### Author Response · Authors · 2021-08-10
> **Small changes in the MCMC transition kernel**
>
> *''The use of Monte Carlo to help with an intractable E step is not a new idea, and I think the paper needs more discussion of the related work in this area. For example, see Levine and Casella (2001) (reference below). Ideas from this literature might be helpful for the current paper – e.g. Levine and Casella (2001) recommend updating the Monte Carlo samples from the previous E step via importance sampling, rather than re-running a MCMC method. Still, I don’t know that other work has handled the issue of a high-dimensional parameter space (here, the number of mixture components K is large), which sounds like an important addition over pre-existing work.''*
>
> Indeed, as we discuss in Section 5 (Related work), there are previous instances of applying an MCMC method to deal with the intractable E-step. As you correctly point out above, our main contribution is to advance beyond the exponential family models (Section 5). Please note that we do not re-run the MCMC method (more on this below). We would like to thank you for suggesting the reference. We will include it in Section 5.
>
> *''The hope of the method is that we can effectively approximate the E step by drawing $M\ll K$ samples from the distribution $p(z_i|x_i)$. It seems that in some sense this is automatically impossible – the distribution $p(z_i|x_i)$ has support of size $K$, so we can’t, in general, approximate expectations over this distribution with only $M$ samples. So I think it’s unlikely that, in general, “the chain passes the transition period (the burn-in phase)” as the authors note. Presumably the authors’ method either relies on some regularity conditions (maybe the distribution $p(z_i|x_i)$ is mostly concentrated on a few points?) or maybe performing the M-step with an only approximate objective is fine, or maybe something else is happening. In any case, can the authors clarify with discussion and/or additional experiments what is driving the success of their method?''*
>
> Please note that $M\ll K$ does not mean that we re-run the Markov chain at each iteration, $t$. The chain is connected across the iterations, hence even for $M=1$, we produce a Markov chain, $(z_{i,t})^T_{t=1}$, that can potentially go to infinity by admitting $T\rightarrow\infty$. This has been stated in the fourth paragraph of Section 4.1: ''... at every current iteration $t$, we continue to extend the chain from the point where we left at the previous iteration, $t-1$, by taking ...'' (see also the sixth line of Algorithm 1). This property has the effect that the burn-in phase takes place at the beginning of the iterations, where the generated non-i.i.d. samples---which are inappropriate to estimate $\theta$---are forgotten via the Robbins-Monro step-size schedule (Section 4.2). After the burn-in phase, the samples are approximately i.i.d. and are used to estimate $\theta$.
>
> *''On lines 130-138, I think the authors are proposing to somehow reuse the samples from $p(z_i|x_i)$ between iterations. But the distribution being sampled from is changing from iteration to iteration (i.e. it depends on the parameters $\theta$ estimated from the M-step).''*
>
> As mentioned above, the idea of applying MCMC to approximate the E-step was utilized before in different application contexts and is theoretically well-supported for the exponential family. The convergence of the sequence of parameter estimates, $(\theta_t)^T_{t=1}$, towards a stationary point of the likelihood surface is proven in Theorem 1 of [31] (and the reference therein) and, similarly, Theorem 1 of [32]. The key intuition behind the convergence towards the target distribution can be obtained from the assumption SAEM3' of Theorem 1 in [31], i.e. the step-size $\gamma_t$ has to be small enough to ensure small changes of $\theta_t$, and, by extension, small changes in the MCMC transition kernel (parameterized by $\theta_{t-1}$). We consider these results to be a very strong justification of this approach for the exponential family (which is confirmed experimentally in Section 6.1). Our experiments provide empirical evidence that similar properties can be expected from its extension to non-exponential family models (Sections 6.2 and 6.3).
>
> *''What do the authors mean on line 131 when they talk about samples being “recycled over the iterations”?''*
>
> Computing the estimate $\theta_t$ directly from (8)---with only the current set of samples $z_{i,t}=(z^1_{i,t},\ldots,z^M_{i,t})$ would make the sequence $(\theta_t)^T_{t=1}$ highly varying (even for large $T$). To make the sequence converge, we accumulate the sets $z_{i,\tau}$ from $\tau=1$ to $\tau=t-1$ via $\eta_{k,t-1}$ and $\nu_{k,t-1}$ in (9), i.e. the past values contain the information from the Markov chain of length $(t-1)M$.
>
> *''The authors parameterize the mixing weights $\pi_k$ by a $K$ real-valued parameters $\nu_1,\ldots,\nu_K$, and then set $\pi_k=e^{\nu_k}/\sum^K_{l=1}e^{\nu_l}$. But this transformation leaves a degree of freedom unconstrained (i.e. scaling all $\nu_k$ by the same constant $c$ will leave the $\pi$’s unchanged), meaning there is no unique optimum of the log-likelihood. This seems potentially undesirable, but maybe isn’t a huge deal.''*
>
> Thank you for spotting this. Indeed, we did not observe this to be an issue.
>
> *''For the purposes of considering an algorithm terminated, the authors stop each algorithm when it reaches 95\% of the maximum likelihood value that it will achieve. I have two questions about this. The first is why is this a reasonable metric? Can we be sure that the fitted parameters will not change much as we traverse the last 5\% of the loss? Second, this doesn’t necessarily seem like a good way to assess convergence of practical algorithms. In particular, the only way to define this stopping criteria is to first run the algorithm to termination so that we know what the final log likelihood value is, and then go back and stop it 95\% of the way there.''*
>
> This rule is used only to define a common point at which to compare the relative speed of convergence of various tested algorithms. It was not meant as a stopping rule.
>
> *''Can the authors argue that this does not bias the results at all? For example, it could be that it takes much longer for their algorithm to traverse the final 5\% of the likelihood than other algorithms.''*
>
> We did not observe this in the experiments. We expect this to be a rare situation. Note, e.g., from Figure 1 that the curve bends rather sharply, and it is commonly the case. Different criteria choosing a fixed point in the bend of the curve will yield very similar results.
>
> *''The authors choose a particular annealing schedule for use across all algorithms. How did they choose this schedule?''*
>
> We choose the deterministic anti-annealing schedule based on the arguments in [43]. The parameters of the annealing are specified in Section 6.1.
>
> *''Does it definitely not bias the results in favor of one algorithm or another?''*
>
> We did not observe any significant impact on the differences between the methods (Table 1) with and without the annealing. We demonstrate this in Section 9.1 of the supplementary material.
>
> *''Are all the experiments done with multiple random restarts for each method? Figure 1 says that “the results are averaged over five repetitions”, but it’s not clear to me how these repetitions are done. Are these five different random initializations? Or five different runs from the same initialization? Each curve in Figure 1 looks like the run from a single random seed, so it seems to be the latter. I think there should be multiple random seeds used so that we know that, for example, TSAEM (the fastest algorithm to reach its optimum in Figure 1) isn’t just getting unluckily stuck in a local optimum in Figure 1.''*
>
> We use different random initializations as described in Section 6.1. To ensure a fair comparison, the same initial values are set for all algorithms in Table 1. We will make this more clear in the text.
>
> *''Why isn’t MHSAEM-TF used in the experiment in Section 6.2? The authors state that it is due to the “favorable computational complexity and simplicity” of MHSAEM-U, but it seems like in Figure 2 the U and TF versions of these algorithms have just about the same performance.''*
>
> In Section 9.3 of the supplementary material, we demonstrate that the TF-proposal is more efficient than the U-proposal for $K=10$. However, as you correctly note, and as we discuss in Section 9.3, the TF-proposal and U-proposal have similar performance for $K=100$ (Figure 2 of the main paper). This reveals that the TF-proposal loses its efficiency for high $K$, which is also the reason we did not use it in Section 6.2. Moreover, TF is much more demanding to implement for deep mixture models such as SPTN, while U is very simple. Nonetheless, even the less efficient U-proposal delivers a substantial speed-up for this model.
>
> *''Similarly to point \# 3, are the results in Table 2 run over multiple random restarts? If so, there should be standard errors reported so that we know how significant the differences are.''*
>
> Yes. Thank you for reminding this to us. We will add the standard deviations.
>
> *''In their paper checklist, the authors write [N/A] for whether their datasets were obtained with consent and whether their datasets contain personally identifiable information or offensive content. Their datasets were taken from the UCI repository, so presumably neither of these things is true, but that doesn't mean it isn't relevant to this paper. I think a quick appendix about the data used would resolve this issue.''*
>
> All the used datasets are tabular data without any string information or personal id numbers so we consider them unoffensive and anonymous. We will add a note to the appendix.

---

> > ### Comment · Reviewer_iZc8 · 2021-08-11
> > **Clarification**
> >
> > Thank you for the detailed replies! I have a few remaining concerns that I was hoping to get some clarifications on:
> >
> > **On the description of the proposed MCMC algorithm**
> >  If I'm understanding correctly, the idea is that the distributions $p_\theta (z | x)$ will change slowly enough as $\theta$ changes so that using the previous iterations samples as a starting point for the new iteration will actually produce a MCMC algorithm converging to the correct distribution. And this happens at the right rate so that $\theta$ can also converge to an actual optimum of the loss. This is a pretty involved argument, and I don't think it was really mentioned in the paper. I think there should be some significant clarification of the description of this algorithm, as it looks like all four reviewers were confused by this point.
> >
> > **On the justification of the proposed MCMC algorithm**
> > The authors note that theorems from [31,32] justify this approach. If this is the case, these should be cited, rather than [52], which is currently used to justify the approach (line 120). Further, it seems like there are a lot of regularity conditions assumed in these papers. Do these hold in the situations the authors are considering? I think the paper needs some significant discussion of these theorems if they're going to be used to justify an important part of the authors' approach. Do the authors see this as a small revision?
> >
> > **On the use of the 95% likelihood for evaluation**. The authors note that this "was not meant as a stopping rule." I would point out that what these experiments are evaluating is the relative performance of the algorithms with this used as a stopping rule. Could one not evaluate the algorithms in the way they will be used in practice, say, with a stopping rule of "terminate when the loss change between iterations is $\leq \delta$, for some small $\delta$"? This more realistic stopping rule looks like it could change some of the qualitative conclusions of Figure 1 -- the purple curve looks like it would terminate almost immediately with this more realistic rule, whereas the yellow and brown would take an extra couple of minutes. It looks like this might double the runtime of the MHSAEM algorithms, putting them behind SSAEM (in terms of runtime, at least). This makes me a little concerned about what might happen to later results.

---

> > > ### Author Response · Authors · 2021-08-18
> > > **Re: Clarification**
> > >
> > > *''If I'm understanding correctly, the idea is that the distributions $p_\theta(z|x)$ will change slowly enough as $\theta$ changes so that using the previous iterations samples as a starting point for the new iteration will actually produce a MCMC algorithm converging to the correct distribution. And this happens at the right rate so that $\theta$ can also converge to an actual optimum of the loss.''*
> > >
> > > Yes, this is the case.
> > >
> > > *''This is a pretty involved argument, and I don't think it was really mentioned in the paper. I think there should be some significant clarification of the description of this algorithm, as it looks like all four reviewers were confused by this point.''*
> > >
> > > We agree that we should have done a better job at explaining this idea. Thanks to you, and the other reviewers, we have a clear idea how to improve our presentation.
> > >
> > > *''The authors note that theorems from [31,32] justify this approach. If this is the case, these should be cited, rather than [52], which is currently used to justify the approach (line 120). Further, it seems like there are a lot of regularity conditions assumed in these papers. Do these hold in the situations the authors are considering? I think the paper needs some significant discussion of these theorems if they're going to be used to justify an important part of the authors' approach. Do the authors see this as a small revision?''*
> > >
> > > We agree that references [31,32] are really more appropriate. Perhaps, even better methodology, which is applicable to prove the convergence of our approach, was proposed in [A]. We propose to improve our motivation in the light of discussion with the reviewers. However, the primary purpose of our paper was not a theoretical study but demonstration of the validity of this approach (and its extension beyond the exponential family) via a thorough experimental study.
> > >
> > > [A] C. Andrieu, E. Moulines, P. Priouret, ``Stability of stochastic approximation under verifiable conditions'', SIAM Journal on control and optimization 44.1:283-312, 2005.
> > >
> > > Regarding the regularity assumptions: The joint distribution, $p_\theta(z,x)$, of the Gaussian mixture model (Section 6.1) is a member of the exponential family. Thus, we satisfy the conditions of Theorem 1 in [31]. A precise check of the conditions of Theorem 1 [A] for the models adopted in Sections 6.2 and 6.3. is a difficult task. We did not validated these conditions, but the experimental results suggest that they may hold.
> > >
> > > *''The authors note that this "was not meant as a stopping rule." I would point out that what these experiments are evaluating is the relative performance of the algorithms with this used as a stopping rule. Could one not evaluate the algorithms in the way they will be used in practice, say, with a stopping rule of "terminate when the loss change between iterations is $\leq\delta$, for some small $\delta$"?''*
> > >
> > > We believe that evaluation of the algorithm using the $\delta$-based stopping rule will introduce another degree of freedom to consider in experimental evaluation since different stopping rules may have different effect for different methods. For example, optimal stopping rules for stochastic algorithms should use a kind of filtering of the fluctuations, the exact form of which may affect its effectiveness, introducing unfair conditions. The proposed rule is an idealized (oracle-like) stopping rule that affects all algorithms in the same way. We agree that we thus measure a rather impractical convergence properties, but it quantifies our main interest, i.e. the convergence of the method. A practically relevant stopping rule may be designed as a next step.
> > >
> > > *''This more realistic stopping rule looks like it could change some of the qualitative conclusions of Figure 1 -- the purple curve looks like it would terminate almost immediately with this more realistic rule, whereas the yellow and brown would take an extra couple of minutes. It looks like this might double the runtime of the MHSAEM algorithms, putting them behind SSAEM (in terms of runtime, at least). This makes me a little concerned about what might happen to later results.''*
> > >
> > > Indeed, the SSAEM algorithm can compete with the MHSAEM algorithm in terms of the run-time, as it is also a techniques for reducing the number of evaluated components. However, as presented by Figure 1, it is not as close to the exact log-likelihood as the MHSAEM algorithm. We observed this behaviour in various experiments. We believe that the most important feature to compare among the algorithms in Table 1 is to achieve the lowest fitting error in the shortest run-time. Given our objective way of assessing this trade-off (for the reasons explained above), Figure 2 demonstrates that the MHSAEM algorithm outperforms the other algorithms in Table 1.

---

> > > > ### Author Response · Authors · 2021-08-27
> > > > **Re: Clarification**
> > > >
> > > > Please, see our reply to the following comment: https://openreview.net/forum?id=ki4eJ1fSJNq&noteId=9BIgwv3grCJ

---

### Official Review · Reviewer_nhhz · 2021-07-16

**Rating:** 4
**Confidence:** 3

**Summary:**

This paper proposes a stochastic approximation to the E-step of the EM algorithm to reduce the computational cost of fitting mixture models when the number of components $K$ is large. Specifically, the paper proposes using $M << K$ samples of the latent variables $z$ from a Metropolis-Hasting (MH) sampler to approximate the E-step. This lets us only use $M$ likelihood evaluations instead of $K$ for each observation in the E-step. The paper compares there proposed method against other EM-based methods on Gaussian mixture models on synthetic data, and sum product  transform networks and non-volume preserving flows on 19 UCI datasets.

**Limitations And Societal Impact:**

N/A.

**Main Review:**

This paper should be rejected.

Originality: the combination of MH-sampler and EM appears new, specifically only sampling $M << K$ points to avoid evaluating the likelihood $K$ times for each point. The paper focus primarily on EM methods, but does not discuss comparisons with variational inference (both EM and VI are variational methods that maximize the ELBO).

Quality: the paper is mostly correct. I have a few small concerns:
For Section 4.1, initializing MH sampler $z_i$ from the last sampled $z_i$ (to avoid using a burnin) seems reasonable, but for large N relative to minibatch size B, isn't it possible the last sampled $z_i$ could be very stale (if $i$ hasn't been selected in a minibatch for a long time)?
For Section 6.1 line 227, doesn't the GMM have an analytic closed form solution for $\theta$ given $x,z$? Why resort to gradient based optimization?
For Section 6.3. I do not know if a large $\Delta \mathcal{L}^{test}$ is meaningful or not. Especially for large $K$. How do you interpret this practically? And for models with $K=16384$ do we expect convergence given the # iteration, $M$ and $N$?
In the conclusion line 316, MH when *run to convergence* ensures sampling from the proper posterior, here we explicitly are only taking $M = 2$ steps. How does this approximation affect the accuracy vs runtime tradeoff? For more complex models, where the runtime benefit of avoiding likelihood computations is larger, but this approximation is less accurate does this algorithm help or hurt?

Clarity: The paper's writing and organization is okay, but the paper could use additional editing to improve grammar + use more professional/less casual tone. For example: the paragraph around line 77.

Significance: Why are we interested in fitting mixture models with large $K$?

Other notes:
* line 49: mdoels -> models
* order of $TKND$ in $O(TKND)$ isn't consistent throughout
* not necessary, but would be nice to include comparisons against gibbs sampling or VI to make the experiments stronger.

**Time Spent Reviewing:**

3

---

> ### Author Response · Authors · 2021-08-10
> **Comparison with variational inference**
>
> *''Originality: the combination of MH-sampler and EM appears new, specifically only sampling $M\ll K$ points to avoid evaluating the likelihood $K$ times for each point. The paper focus primarily on EM methods, but does not discuss comparisons with variational inference (both EM and VI are variational methods that maximize the ELBO).''*
>
> We would like to thank you for your valuable feedback. Indeed, the EM algorithm is a variational method, as we point out in Section 1 (Introduction) and Section 3 (The generalized MHSAEM algorithm). All methods in Table 1---which we compare in Section 6 (Experiments)---are different variants of the EM algorithm, each having a different approximation of ELBO. The paper thus contains a rich comparison of a class of variational methods which all follow from the common basis (the ELBO of the full EM algorithm). We considered comparing other variational methods, but they would lack such a tight methodological connection and would involve more substantial differences, making it hard to argue for the fairness of experimental outputs. For these reasons, we believe that our current comparison is principled and well-chosen.
>
> *''Quality: the paper is mostly correct. I have a few small concerns: For Section 4.1, initializing MH sampler $z_i$ from the last sampled $z_i$ (to avoid using a burnin) seems reasonable...''*
>
> Please note that we do not try to avoid the burn-in. The key idea behind using $z^M_{i,t-1}$ in the next iteration, $t$, is to connect the chain over the iterations (line 6 of Algorithm 1). Hence, even for $M=1$, we produce a full Markov chain, $(z_{i,t})^T_{t=1}$. This property has the effect that the burn-in phase takes place at the beginning of the iterations, where the generated non-i.i.d. samples---which are inappropriate to estimate $\theta$---are forgotten via the Robbins-Monro step-size schedule (Section 4.2). More specifically, in the burn-in, we set $\gamma_t$ sufficiently small to avoid learning improper values of $\theta$. After the burn-in phase, the samples are approximately i.i.d. and are used to estimate $\theta$. Thanks to you, we see that we have to improve our explanation in this respect.
>
> *''...but for large $N$ relative to minibatch size $B$, isn't it possible the last sampled $z_i$ could be very stale (if $i$ hasn't been selected in a minibatch for a long time)?''*
>
> Your intuition is correct. This can temporarily lower the acceptance ratio for a given $i$ but should be quickly compensated for if a sufficiently informed and efficient proposal distribution is used.
>
> *''For Section 6.1 line 227, doesn't the GMM have an analytic closed form solution for $\theta$ given $x,z$? Why resort to gradient based optimization?''*
>
> The comment on line 227 states that we use, indeed, the sufficient statistics in the experiments of Section 6.1 (and only there) to be comparable to the rest of the methods in Table 1 (which are unable to deal with non-exponential family models). We will make the assertion more precise.
>
> *''For Section 6.3. I do not know if a large $\Delta\mathcal{L}^{\text{test}}$ is meaningful or not. Especially for large $K$. How do you interpret this practically?''*
>
> We do not use $\Delta\mathcal{L}^{\text{test}}$ in our text. Could you, please, clarify your question?
>
> *''And for models with $K=16384$ do we expect convergence given the \# iteration, $M$ and $N$?''*
>
> Please note that, in Section 6.2, we state that the number of components of an SPTN after its conversion into a mixture model (SPTN expansion) is $K=s^l$. However, this value serves only to make the models in Table 2 comparable for the reader. The parameter updates are not performed in the expanded form but in sub-trees. In our implementation, we sample $M=1$ sub-tree (Section 6.2) which contains several nodes of the network (see, e.g., Figure 1 and Definition 3 in [A]). Therefore, multiple components from $K=16384$ are updated since parameters of one sub-tree in our architecture influence $s^{(l-1)}$ components in the expanded form. We found that the number of iterations, $T=20k$, is sufficient in the experiments.
>
> [A] P. Clavier, O. Bouaziz, G. Nuel, 'Gaussian Sum-Product Networks Learning in the Presence of Interval Censored Data', In Proceedings of the 10th International Conference on Probabilistic Graphical Models, PMLR 138:125-136, 2020.
>
> *''In the conclusion line 316, MH when run to convergence ensures sampling from the proper posterior, here we explicitly are only taking $M=2$ steps. How does this approximation affect the accuracy vs runtime tradeoff?''*
>
> We have run experiments with various values of $M$, see right-most subfigure of Figure\ 2 of the main text where the tradeoff is visualized for Gaussian mixture models. It is true that there seems to be a an optimal value of $M$ for this task, however, it may be significantly problem-dependent. We have used only $M=1$ for deep mixtures in Section 6.2 and Section 6.3 which was found sufficient due to connecting the chain as mentioned above.
>
> *''For more complex models, where the runtime benefit of avoiding likelihood computations is larger, but this approximation is less accurate does this algorithm help or hurt?''*
>
> Our experiments suggest that our algorithm helps since it often reaches the best log-likelihood. We conjecture that this is due to the fact that stochastic sampling of components not only approximates the E-step but also provides an additional ''exploration" of the space, allowing to escape from a poor local minimum.
>
> *''Clarity: The paper's writing and organization is okay, but the paper could use additional editing to improve grammar + use more professional/less casual tone. For example: the paragraph around line 77.''*
>
> Thank you. We will sharpen our assertion there.
>
> *''Significance: Why are we interested in fitting mixture models with large $K$?''*
>
> Our main motivation comes from the training of deep mixture models such as SPNs and SPTNs, where the number of components grows exponentially with the number of layers. The conventional  SGD algorithm tends to be slow to learn such models for tasks such as density estimation, in which the variational auto-encoders and generative adversarial networks are typically faster. Thus mixture models are perceived as inferior and receive less attention from the community. The speed-up attainable with our method can make SPTNs more competitive for tasks like density estimation.
>
> *''order of $TKND$ in $\mathcal{O}(TKND)$ isn't consistent throughout''*
>
> Thank you for spotting this. We will correct our text accordingly.
>
> *''not necessary, but would be nice to include comparisons against gibbs sampling or VI to make the experiments stronger.''*
>
> As indicated above, we recognize several principal differences between these (presumably Bayesian) methods and our maximum-likelihood method. The ideas from the algorithms in Table 1 could potentially be utilized in the methods you suggest, creating two new classes of techniques. Comparing methods in each of these classes individually is fair. However, this is not the case for different groups of methods, say, methods based on Gibbs sampling and maximum-likelihood methods. We will consider this as a possible direction for future work.

---

> > ### Comment · Reviewer_nhhz · 2021-08-10
> > **Clarifications**
> >
> > 1. *MCMC going to $\infty$.*
> > In the response to all reviews you refer to the MCMC chain as going to $\infty$ (even when $M = 1$) because the chain is "connected across iterations". But since the parameter $\theta$ changes between iterations, doesn't the Markov chain kernel change on every step? That is, the MH step accept-reject probabilities change when $\theta$ changes? And as a result the chain $z_{i,t}$ is not a "proper" Markov chain (as the stationary distribution of the kernel changes on every step $t$).
> > If the model is "smooth" enough or the step size $\gamma_t$ is small enough, such that $\theta_t$ and the Markov chain kernel change slowly enough, then it's reasonable to believe the distribution of $z_{i,t}^M$ might be approximately close to an equivalent sample $z_{i,t}^{\infty}$. But it's not clear how small the change in $\theta_t$ would need to be for this to be an accurate approximation (especially in the beginning).
> > And the matter gets worse as $z_i$ is only infrequently updated (i.e. the effective change in $\theta_t$ between MH steps gets larger). Can you clarify this for me?
> >
> > 2. For section 6.2 and 6.3: $\Delta \mathcal{L}^{test}$, I was refering to comparing the $\mathcal{L}^{test}$ values in Table 2. You show improvement on the test loglikelihood using your method, but what does that practically mean?

---

> > > ### Author Response · Authors · 2021-08-18
> > > **Re: Clarifications**
> > >
> > > *''MCMC going to $\infty$. In the response to all reviews you refer to the MCMC chain as going to $\infty$ (even when $M=1$) because the chain is ''connected across iterations". But since the parameter $\theta$ changes between iterations, doesn't the Markov chain kernel change on every step? That is, the MH step accept-reject probabilities change when $\theta$ changes? And as a result the chain $z_{i,t}$ is not a ''proper'' Markov chain (as the stationary distribution of the kernel changes on every step $t$).''*
> > >
> > > Indeed, $z_{i,t}$ is a time non-homogeneous Markov chain. The common technique for dealing with such chains in the context of stochastic approximation is to transform a non-homogeneous chain into a homogeneous one to prove convergence [A].
> > >
> > > [A] C. Andrieu, E. Moulines, P. Priouret, ``Stability of stochastic approximation under verifiable conditions'', SIAM Journal on control and optimization 44.1:283-312, 2005.
> > >
> > > *''If the model is ''smooth" enough or the step size $\gamma_t$ is small enough, such that $\theta_t$ and the Markov chain kernel change slowly enough, then it's reasonable to believe the distribution of $z^M_{i,t}$ might be approximately close to an equivalent sample $z^\infty_{i,t}$. But it's not clear how small the change in $\theta_t$ would need to be for this to be an accurate approximation (especially in the beginning).''*
> > >
> > > Since this is still the material discussed in [31], we will try to refer to particular places in it. We review the point 2 of assumption SAEM3'. For any compact subset $V$ of $\Theta$, there is the Lipschitz constant which sets the admissible bound on the change between two consecutive values of the parameters, $(\theta,\theta')\in V^2$,
> > > $$
> > >     \underset{(x,y)\in\mathcal{E}^2}{\operatorname{sup}}|\Pi_{\theta}(x,y)-\Pi_{\theta'}(x,y)|\leq L|\theta-\theta'|
> > >     ,
> > > $$
> > > where $\Pi_{\theta}(x,y)$ is the transition probability distribution on $\mathcal{E}$, defining an MCMC algorithm (the MH algorithm in our case, with $\theta' \equiv \theta_t$ and $\theta \equiv \theta_{t-1}$). Theoretically, one can find an analytical expression for $L$. This is, however, a rather complicated task, which is usually not undertaken in practice. The change between the two values is often restricted by appropriately choosing the step-size sequence $(\gamma_t)^T_{t=1}$ which has to satisfy the Robbins-Monro conditions (Section 4.2). The choice of a specific schedule is done heuristically, but in general, as long as $\theta$ belongs into a compact subset of $\Theta$, the change can be relatively big. Rapid changes of $\theta$ typically take place during the initial iterations of the algorithm, where we require fast exploration of the likelihood surface. After the initial iterations, a decreasing (or constant with small-enough $\gamma_t$) sequence is applied to ensure convergence close to a local optimum, see, e.g., Section 4.1 of [31]. Once again, we assume that the same mechanism will apply to our case.
> > >
> > > *''And the matter gets worse as $z_{i}$ is only infrequently updated (i.e. the effective change in $\theta_t$ between MH steps gets larger). Can you clarify this for me?''*
> > >
> > > You are right that for $B\ll N$ the change in $\theta$ may become larger. However, the condition with Lipschitz constant above has to hold for any difference between $\theta-\theta'$. Moreover, this situation may be improved by increasing $M$.
> > >
> > > *''For section 6.2 and 6.3: $\Delta\mathcal{L}^{\text{test}}$, I was refering to comparing the $\mathcal{L}^{\text{test}}$ values in Table 2. You show improvement on the test loglikelihood using your method, but what does that practically mean?''*
> > >
> > > The aim of the experiment is to estimate the density function from given samples. The test log-likelihood is a standard application measure of quality in this task to verify also the tendency of the method to over-fitting. The absolute value of the log-likelihood (or its difference) is indeed hard to interpret and problem-specific. The typical use is for ranking of methods, a method with higher value of the log-likelihood is considered to be better. That is why we summarize the results with ranking statistics at the bottom line.

---

> > > > ### Comment · Reviewer_nhhz · 2021-08-19
> > > > **Re: Re: clarification**
> > > >
> > > > Thank you for the clarifications.
> > > >
> > > > This discussion about the approximated convergence is important and should be included (at least alluded to in the main paper and discussed in the supplement). Ultimately, if we only care about $\theta$, the stochastic gradients for the M-step do not necessarily need to be perfect/unbiased as long as the bias is controlled (see Nodedal and Wright 2006).
> > > >
> > > > Perhaps a toy experiment showing the bias / relative error of the gradient estimator over training would help show this empirically? That is, periodically during training, we could estimate the bias of the gradient (9) at step $t$ by computing the exact gradient (say in the GMM case) and multiple instances of your MHSAEM gradient (9) over different random subsets $I$ starting from the same $z_{\cdot,t}$? Maybe fore multiple paths?
> > > >
> > > > Thank you for the clarification on the experimental evaluation. I agree that the interpretation of test log-likelihood (and rank for that matter) is likely to be problem/data-set specific. The practical impact on these specific datasets is poor, but at least shows the computational speed-up of MHSAEM for large K on non-synthetic data.

---

> > > > > ### Author Response · Authors · 2021-08-27
> > > > > **Re: Re: Re: Clarifications**
> > > > >
> > > > > We are very grateful to you for suggesting this experiment. Here, we illustrate the convergence on a toy GMM example evaluating the bias in the following way: $\text{bias}=||\nabla_{\theta}Q_{\text{SAEM}}-\nabla_{\theta}\hat{Q}||$, where $\nabla_{\theta}Q_{\text{SAEM}}$ is the exact gradient of the SAEM algorithm which computes the full expected value, $\nabla_{\theta}\hat{Q}$ is the approximate gradient of the MHSAEM algorithm which computes the expected value as shown in (8), and $||\cdot||$ is the $l_2$-norm. We also include the angle between the gradients, $\delta=\operatorname{acos}(\nabla_{\theta}Q_{\text{SAEM}}\cdot\nabla_{\theta}\hat{Q}/{||\nabla_{\theta}Q_{\text{SAEM}}||}{||\nabla_{\theta}\hat{Q}||})$. The result is averaged over 20 repetitions using a single dataset generated with $(D,K,N,\omega,B,M,T)=(2,10,1000,0.001,100,1,1000)$. We use the following step-size sequence: $(0.8t^{-0.5})^T_{t=1}$. The experiment demonstrates that, even for $M=1$, the bias goes to zero as the iterations increase (the convergence is thus achieved over multiple iterations, $t$). This empirically confirms that the E-step is asymptotically unbiased, which implies convergence of the algorithm and the validity of the method. While formal proof of convergence of our method is not yet available, we expect it to be analogical to that in [31].
> > > > >
> > > > > Please, use the following link: https://picc.io/tSizQZ7.png
> > > > >
> > > > > ![](https://picc.io/tSizQZ7.png)

---

### Official Review · Reviewer_n8QC · 2021-07-17

**Rating:** 4
**Confidence:** 3

**Summary:**

This paper proposes a stochastic EM algorithm. Specifically, the authors use a minibatch to approximate the data and use samples obtained from Metropolis-Hasting (MH) to approximate the likelihood component per datum. The proposed method is tested on a synthetic dataset and several UCI datasets.

**Limitations And Societal Impact:**

The paper does not contain this part.

**Main Review:**

The main contribution of this paper is the proposed likelihood component approximation using MH, as using minibatch to approximate the dataset is pretty common in ML. The empirical results of the proposed method seem promising, but I have some concerns about the methodology.

It is known that MCMC needs a burn-in phase in order to produce samples from the correct distribution. The proposed algorithm only runs M steps where M needs to be small enough to save computation. It is hard to believe the samples are from the right distribution. Moreover, the algorithm collects consecutive samples thus the samples will be highly correlated. What is the acceptance probability of MH? It seems that these will introduce bias to the approximation. Besides introducing bias, how is the variance of this approximation?

In summary, the idea is reasonable and there seems empirical performance. But the methodology is not very sound.


**Time Spent Reviewing:**

5

---

> ### Author Response · Authors · 2021-08-10
> **Burn-in and convergence to the target distribution**
>
> *''It is known that MCMC needs a burn-in phase in order to produce samples from the correct distribution. The proposed algorithm only runs M steps where M needs to be small enough to save computation. It is hard to believe the samples are from the right distribution. Moreover, the algorithm collects consecutive samples thus the samples will be highly correlated.''*
>
> Please note that we do not run the Markov chain only within a single iteration, $t$. The chain is connected across the iterations, hence even for $M=1$, we produce a Markov chain, $(z_{i,t})^T_{t=1}$, that can potentially go to infinity by admitting $T\rightarrow\infty$. This has been stated in the fourth paragraph of Section 4.1: ''... at every current iteration $t$, we continue to extend the chain from the point where we left at the previous iteration, $t-1$, by taking ...'' (see also the sixth line of Algorithm 1). This property has the effect that the burn-in phase takes place at the beginning of the iterations, where the highly correlated, non-i.i.d., samples---which are, indeed, inappropriate to estimate $\theta$---are forgotten via the Robbins-Monro step-size schedule (Section 4.2). After the burn-in phase, the samples are approximately i.i.d. and are used to estimate $\theta$. The convergence towards the target distribution is discussed below.
>
> *''What is the acceptance probability of MH?''*
>
> Acceptance probabilities are discussed in Section 9.3 of the supplementary material and displayed in Figure 7. Similar trends were observed for the remaining experiments.
>
> *''It seems that these will introduce bias to the approximation. Besides introducing bias, how is the variance of this approximation?''*
>
> The idea of applying MCMC to approximate the E-step was utilized before in different application contexts and is theoretically well-supported for the exponential family. The convergence of the sequence of parameter estimates, $(\theta_t)^T_{t=1}$, towards a stationary point of the likelihood surface for this family is proven in Theorem 1 of [31] (and the reference therein) and, similarly, Theorem 1 of [32], containing remarks on that the sequence is asymptotically normal at the standard rate $t^{-0.5}$, i.e. the variance approaches to zero. The key intuition behind the convergence towards the target distribution can be obtained from the assumption SAEM3' of Theorem 1 in [31], i.e. the step-size $\gamma_t$ has to be small enough to ensure small changes of $\theta_t$, and, by extension, small changes in the MCMC transition kernel (parameterized by $\theta_{t-1}$). We consider these results to be a very strong justification of this approach for the exponential family (which is confirmed experimentally in Section 6.1). Our experiments provide empirical evidence that similar properties can be expected from its application to non-exponential family models (Sections 6.2 and 6.3).
>
> *''Limitations And Societal Impact: The paper does not contain this part.''*
>
> Section 7 (Conclusion) discusses the limit in the speed of convergence due to the use of the uniform proposal distribution. Section 8 (Broader impact statement) considers the environmental and economic impact.

---

> > ### Author Response · Authors · 2021-08-27
> > **Clarification**
> >
> > Please, see our reply to the following comment: https://openreview.net/forum?id=ki4eJ1fSJNq&noteId=9BIgwv3grCJ

---

### Decision · Program_Chairs · 2021-09-28

**Decision:**

Reject

**Comment:**

Reviewers are intrigued by the proposed approach to learning mixture models, but voice a number of substantial concerns related to both the clarity and correctness of the technical material on Monte Carlo approximations.  The author response was considered, but for some reviewers, it raised more questions than it answered.  After some discussion, the consensus was that very-major revisions are needed to clearly explain and justify the proposed method, and that such changes require a full review at some future conference.

**Consistency Experiment:**

NeurIPS has a long history of experimentation. In 2014, NeurIPS ran an experiment in which 10% of submissions were reviewed by two independent committees to quantify the randomness in the review process. This year, we repeated a variant of this experiment to see how the quality of the review process has changed over time.  This paper was part of the experiment and was therefore assigned to two committees (consisting of reviewers, an Area Chair, and a Senior Area Chair) that reached independent decisions.  If both committees made the same recommendation, this recommendation was followed. If a single committee recommended acceptance, the paper was accepted (with the exception of a few cases in which the other committee identified what we considered a fatal flaw, e.g., an error in a key result).

Both committees reached the same decision: **Reject**

The other committee assigned to the paper recommended **Reject**.  You can find the other set of reviews, along with any follow up discussion with the authors here:
https://openreview.net/forum?id=ZS394D3djsg